# AWI-CM3 coupled climate model: Description and evaluation experiments for a prototype post-CMIP6 model

Jan Streffing[1,2], Dmitry Sidorenko[1], Tido Semmler[1], Lorenzo Zampieri[3], Patrick Scholz[1], Miguel Andrés-Martínez[1], Nikolay Koldunov[1], Thomas Rackow[4,1], Joakim Kjellsson[5], Helge Goessling[1], Marylou Athanase[1], Qiang Wang[1], Jan Hegewald[1], Dmitry V. Sein[1,6], Longjiang Mu[7,1], Uwe Fladrich[8], Dirk Barbi[9,1], Paul Gierz[1], Sergey Danilov[1,2], Stephan Juricke[1,2], Gerrit Lohmann[1,10], and Thomas Jung[1,10]

[1]Alfred Wegener Institute, Helmholtz Centre for Polar and Marine Research, Am Handelshafen 12, 27570 Bremerhaven, Germany
[2]Jacobs University Bremen, Campus Ring 1, 28759 Bremen, Germany
[3]National Center for Atmospheric Research, 1850 Table Mesa Dr, Boulder, CO 80305, United States of America
[4]European Centre for Medium-Range Weather Forecasts, Robert-Schuman-Platz 3, 53175 Bonn, Germany
[5]GEOMAR Helmholtz Centre for Ocean Research Kiel, Wischhofstraße 1-3, 24148 Kiel, Germany
[6]Shirshov Institute of Oceanology, RAS, Moscow, Russia
[7]Pilot National Laboratory for Marine Science and Technology, Qingdao, China
[8]Swedish Meteorological and Hydrological Institute, Folkborgsvägen 17, SE-60176 Norrköping, Sweden
[9]Rhenish Friedrich Wilhelm University of Bonn, Regina-Pacis-Weg 3, 53113 Bonn, Germany
[10]University of Bremen, Bibliothekstraße 1, 28359 Bremen, Germany

**Correspondence:** Jan Streffing (Jan.Streffing@awi.de), Dmitry Sidorenko (Dmitry.Sidorenko@awi.de)

**Abstract.** We developed a new version of the Alfred Wegener Institute Climate Model (AWI-CM3), which has higher skills in representing the observed climatology and better computational efficiency than its predecessors. Its ocean component FESOM2 has the multi-resolution functionality typical for unstructured-mesh models, while still featuring a scalability and efficiency similar to regular-grid models. The atmospheric component OpenIFS (CY43R3) enables the use of latest developments in the numerical weather prediction community in climate sciences. In this paper we describe the coupling of the model components and evaluate the model performance on a variable resolution (25–125 km) ocean mesh and a 61 km atmosphere grid, which serves as a reference and starting point for other on-going research activities with AWI-CM3. This includes the exploration of high and variable resolution, the development of a full Earth System Model as well as the creation of a new sea ice prediction system. At this early development stage and with the given coarse to medium resolutions, the model already features above CMIP6-average skills in representing the climatology and competitive model throughput. Finally we identify remaining biases and suggest further improvements to be made to the model.

## 1 Introduction

The evolution of coupled climate models between phases of the Coupled Model Intercomparison Project (CMIP) is advancing our ability to simulate the Earth's climate and to quantify humankind's past and future impact.

The Alfred Wegener Institute (AWI) contributed to CMIP6 with the Atmosphere-Ocean General Circulation Model (AOGCM) AWI-CM1.1-MR (Semmler et al., 2020) as well as the Earth System Model (ESM) AWI-ESM1.1-LR (Danek et al., 2020), built upon AWI-CM1 through using an additional dynamic vegetation module. The AOGCM version also contributed to the HighResMIP of CMIP6 (Rackow et al., 2022) with AWI-CM1.1-LR and AWI-CM1.1-HR.

Experience from HighResMIP shows that, with respect to model accuracy, high resolution climate models are about one
CMIP generation ahead of their standard resolution counterparts (Bock et al., 2020). Moreover, atmospheric modeling studies indicate that a number of key processes ranging from orographic drag (Pithan et al., 2016) to atmospheric blocking (Schiemann et al., 2017; Davini et al., 2017), storm tracks (Willison et al., 2015; Baker et al., 2019), precipitation (van Haren et al., 2015), as well as mean sea level pressure (Hertwig et al., 2015), can be improved by increased horizontal resolution. In many of these cases the operational resolutions of the current NWP systems ( 10km) would be sufficient, but are often not reached by climate
models running more commonly at  100km resolution, due to computational cost and time-to-solution limitations.

While the scientific evaluation of CMIP6 is still ongoing, we turn our attention to lessons learned and begin the development of our next-generation climate model. Our CMIP6 model, AWI-CM1, has reached the limits of scalability, both in the design of its numerical cores, and in the peripheries, such as data structures, and IO schemes. As a first step forward, AWI embarked on a mission to create a FESOM version 2.0 with a finite volume numerical core instead of finite elements (Danilov et al., 2017;
Scholz et al., 2019; Koldunov et al., 2019a). In the vertical dimension the Arbitrary Lagrangian Eulerian (ALE) framework in FESOM2 allows the vertical grids to follow the isopycnals with reduced numerical mixing and to follow bottom topography with improved representation of bottom boundary layers. FESOM2 thus bears resemblance to the MPAS model described by Petersen et al. (2015). Following the development of FESOM2, a new climate model was assembled, coupling this upgraded ocean model with the same atmospheric model as before, ECHAM6 (Sidorenko et al., 2019). This model, dubbed AWI-CM2,
is however practically limited to an atmospheric resolution of about 100 km grid spacing, with an absolute upper limit of 50 km.

Highly relevant atmosphere-ocean coupled processes such as local energy transfer, ocean warm layer formation, and diurnal cycles require not only high resolution in the ocean component, but an atmosphere that can adequately react to the ocean in an eddying regime (Ma et al., 2016; Renault et al., 2016). We therefore couple FESOM2 to the OpenIFS atmospheric model to develop our new AOGCM AWI-CM3. OpenIFS is based on the Integrated Forecasting System (IFS) numerical weather
prediction suite. IFS and OpenIFS are highly scalable and have been used in the Centre of Excellence in Simulation of Weather and Climate in Europe (ESiWACE) (Zeman et al., 2021). They constitute the highest atmospheric resolution contribution to HighResMIP (Haarsma et al., 2016), and have a long history of experimental (Jung et al., 2012) and operational (Malardel et al., 2016) high resolution atmospheric modelling.

While it might be tempting to apply the coupled model at the highest possible spatial resolution right from the start, cost and
time considerations dictate that initial development and evaluation best be done at lower resolution. Furthermore many future applications of AWI-CM3, especially as the basis for a paleoclimate-capable full Earth System Model (ESM), will likely not employ particularly high resolutions, due to long simulation periods and/or a large number of tracers. Finally, conducting this first model development phase at relatively low resolutions enables a fair comparison between this model and the old AWI-CM1.1, as well as other well-established climate models. We therefore present in detail the capabilities and scientific

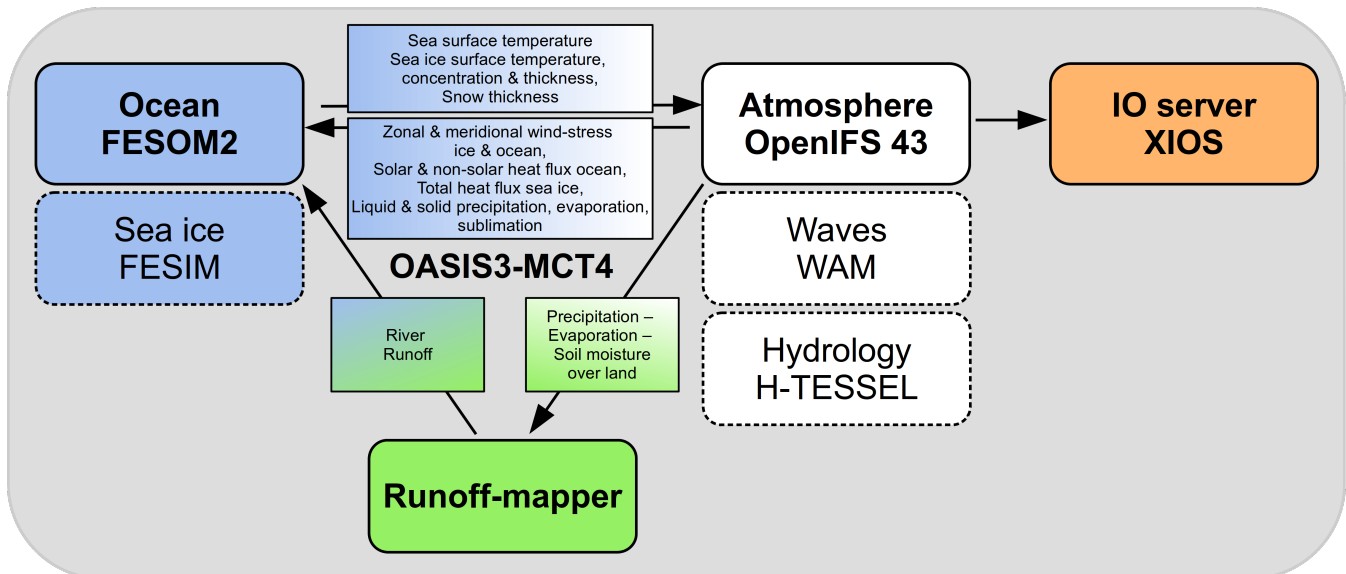

**Figure 1.** Schematic of the coupling between AWI-CM3 model components. Two hourly parallel communication in AWI-CM3 is implemented via the OASIS3-MCT 4.0 library. Heat, mass and momentum fluxes are sent from OpenIFS to FESOM2, while ocean and ice surface state variables are sent back. The precipitation minus evaporation over land is handled with a runoff mapper that generates river runoff at basin discharge points. OpenIFS output is written in parallel with optional online postprocessing via XIOS.

applicability of the lower-resolution AWI-CM3 here, with a glance at its higher-resolution performance. What we consider low resolution for the OpenIFS atmosphere TCo159L91 (61km) is already beyond the practical limits of our previous AWI-CM1 and AWI-CM2 models with ECHAM6 (100 km). The higher resolution simulation that we briefly touch on features a TCo319L137 (31km) atmosphere as well as a 5-27km ocean, making the simulation more finely resolved than the HighResMIP contribution with AWI-CM1.1 HR.

## 2 Model Components

The AWI-CM3 coupled system encompasses two major components with the atmosphere (OpenIFS) and ocean (FESOM2), as well as an auxiliary component, the runoff mapper. The fourth component (XIOS), running in parallel, handles the output from the atmospheric model.

### 2.1 OpenIFS 43R3 Atmosphere

For its atmospheric component AWI-CM3 uses OpenIFS, which is based on ECMWF's Integrated Forecast System (IFS) (ECMWF, 2017a, b, c). With the readily available OpenIFS, ECMWF aims to provide cutting-edge performance from the world

of operational numerical weather forecasting to the research community, while in turn presenting a testbed for developments that can feed back into weather forecasting.

As such, OpenIFS contains the hydrostatic dynamical core, the physical parameterizations, as well as the H-TESSEL hydrology model (Balsamo et al., 2009) and the WAM wave model (Komen et al., 1996) from the ECMWF IFS suite. The two-way OpenIFS-WAM coupling makes the surface roughness calculation dependent on the wave state, which in turn influences the calculation of momentum and sensible heat fluxes. WAM wave fields are currently not directly coupled to FESOM2 for e.g. wave induced vertical mixing calculations. H-TESSEL provides column model type soil moisture computations, while horizontal water transport on land is simulated using a separate runoff-mapper, as shown in figure 1. In contrast to the operational IFS NWP system, the 4D-var data assimilation, the non-hydrostatic core, the adjoint/tangent-linear versions, the Meteo France IO server, and the subroutine level implementation of the NEMO ocean model have been removed from OpenIFS.

The defining feature of the OpenIFS numerical core is its semi-Lagrangian (Ritchie et al., 1995) semi-implicit (Robert et al., 1972) advection scheme (Ritchie, 1987; Ritchie et al., 1995), which allows for advection distances in excess of the Courant–Friedrichs–Lewy (CFL) condition. At the same grid resolution and with the same characteristic fluid velocities this permits OpenIFS simulations to be numerically stable at much longer timesteps than equivalent Eulerian integrations. What the model saves in terms of computing resources can be invested instead into higher model resolution, larger ensembles or longer simulations. We use the cycle 43R3V1 version of OpenIFS released in September 2020, that is based on IFS CY43R3, which constituted the operational NWP system at ECMWF between July 2017 and June 2018. In contrast to ECHAM6, OpenIFS allows for the use of both full and reduced Gaussian grids (Hortal and Simmons, 1991), thus reducing grid cell shape distortion near the poles.

OpenIFS is available at a wide variety of horizontal resolutions, ranging from a TQ21 (626km) toy model to the TCo1279 (9km) operational NWP. Even higher experimental horizontal resolutions require only minor source code changes. Three options exist for the representation of highest (truncation) resolution spherical harmonics in grid point space. The linear truncation TL grids with two gridpoints for the smallest spherical harmonics, the quadratic TQ with three and the cubic octahedral TCo with four grid points (Malardel et al., 2016). Of these, the TCo grids are the most accurate and applicable to coupled climate simulations, as the coupling takes place in grid point space. The vertical resolution choice is much more limited than the horizontal one, as each horizontal resolution is typically paired with one optimal vertical resolution, for which the model parameterizations have been tuned.

The main experiments we present were performed at a resolution of TCo159 (61km) with 91 vertical layers (TCo159L91), with some experiments at TCo319L137 (31km). Further resolutions we successfully tested computationally are TCo95L91 (100km) and TCo639L137 (16km). All grids have a ceiling pressure level of 0.01 hPa, with the L137 grids resolving the vertical space in between more finely than the L91 grids.

## 2.2 FESOM2 Ocean

The ocean dynamics of AWI-CM3 are simulated by the Finite volumE Sea ice Ocean Model (FESOM2), the second version of the global unstructured-mesh ocean model developed at AWI (Danilov et al., 2017).

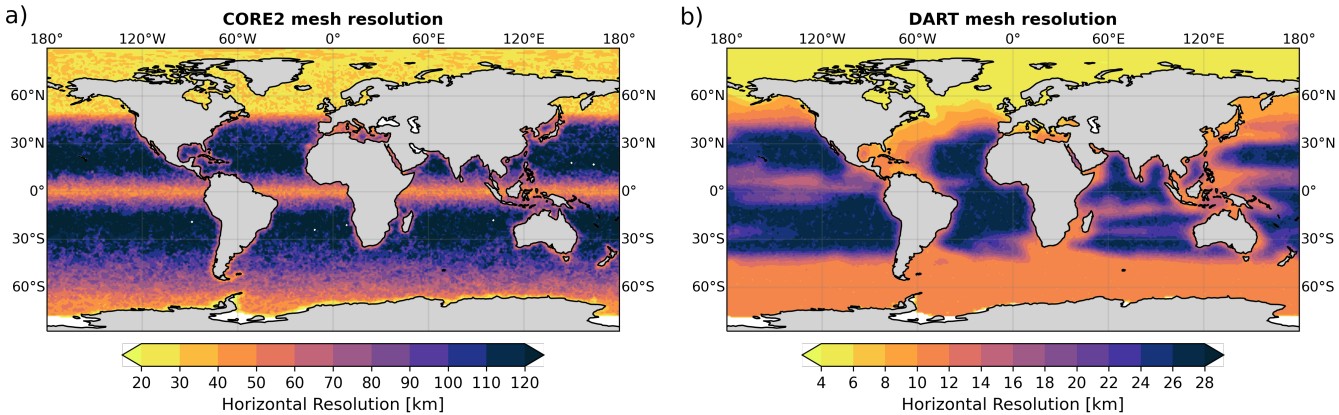

**Figure 2. a)** Horizontal resolution of the CORE2 mesh used for the main simulations presented in this work. **b)** As a) but for the DART mesh with higher resolution, which is based on a mix of local Rossby radius of deformation and local sea surface height variability. The DART mesh is used in Section 5.1 for an outlook of high resolution applications.

FESOM2 is formulated with a finite volume dynamical core using ALE vertical coordinates. FESOM2 is a global unstructured-mesh ocean model that has computational performance comparable to structured-mesh models. Unstructured meshes allow for local mesh refinements without sharp resolution boundaries, as encountered by classical nesting models. In practice, the mesh can be designed to follow the patterns of local sea surface height variability or to scale corresponding to the local Rossby radius of deformation (Sein et al., 2017). FESOM2 contains the embedded FESIM sea ice model (Danilov et al., 2015). For the coupled model presented here, FESIM was modified such that it calculates prognostically the sea ice surface temperature, while in the previous AOGCMs AWI-CM1 and AWI-CM2 this computation was performed in the atmospheric component.

For the experiments presented here we employ a mesh called CORE2, which has about 127k surface nodes, as shown in Koldunov et al. (2019a). The mesh has 47 vertical layers and horizontal resolution varying from 25 to 125 km depending on latitude and distance to coastlines. We tested a second mesh called DART with shorter simulations. The DART mesh has 3.1 million surface nodes with 80 vertical layers and horizontal resolution of 5 to 27km. We show the spacial distribution of horizontal resolution for both meshes in Figure 2. In the vertical, for the first two layers of both meshes have 5 meters resolution, and subsequent layers are in 10 meter intervals till 100 meters depth. Below 100 meters depth the CORE2 vertical resolution decreases downwards more rapidly than the DART resolution.

The low-resolution CORE2 mesh for FESOM2 is not eddy-resolving, and we employ the Gent-McWilliams (GM) parameterization to include the effect of mesoscale eddies on temperature, salinity and tracers (Gent and Mcwilliams, 1990). For all runs presented here, FESOM2 was configured with the "zstar" vertical coordinate, where the total change in sea surface height is distributed over all layers in the vertical, except the partial cell layer at the bottom (Scholz et al., 2019). Such a vertical setup reduces erroneous numerical mixing in the vertical (Adcroft and Campin, 2004). Parameterization of physical vertical mixing is achieved via the K-Profile Parameterization (KPP) scheme after Large et al. (1994). In the Southern Ocean we additionally apply vertical mixing within the Monin–Obukhov length scale calculated based on heat flux, freshwater flux, wind stress, sea

ice concentration and sea ice velocity, as developed by Timmermann and Beckmann (2004) based on Lemke (1987). For the horizontal viscosity, FESOM2 applies the kinematic backscatter scheme of Juricke et al. (2020) with a backscatter coefficient of 1.5. This scheme dissipates kinetic energy on small scales, but reinjects kinetic energy on large scales, resulting in an overall reduced dissipation. Detailed information on the available mixing scheme options in FESOM2 can be found in Scholz et al. (2022). For the experiments shown below, the net evaporation - precipitation - runoff (E-P-R) integrated over the global ocean is forced to 0 at each timestep by subtracting the residual.

The FESIM sea ice model is integrated directly in FESOM2 source code on a module level. The sea ice computations are done on the ocean surface grid and the dynamics use an adaptive elastic-viscous-plastic solver (Kimmritz et al., 2016; Koldunov et al., 2019b). For the thermodynamics FESIM implements a 0-layer scheme after Parkinson and Washington (1979). The standalone ocean model FESOM2 allows for the optional use of the Icepack sea ice thermodynamics module (Hunke et al., 2020; Zampieri et al., 2021), representing a potential future upgrade for AWI-CM3.

Bodies of water that are cut off by land from the world oceans, such as the Caspian Sea and the Great Lakes are not simulated via FESOM2, but are included as lakes via the OpenIFS lake module.

For the low-resolution CORE2 mesh in particular, several ocean basins with narrow outflow channels are not included. These are: the White Sea, Persian Gulf, Black Sea, and the Gulf of Ob. Such narrow inlets on a coarse mesh would reduce the smallest horizontal grid spacing, and thus incur a smaller timestep in order to still fulfill the CFL condition globally. The higher resolution DART mesh does not omit these basins.

## 2.3 Runoff-mapper

In addition to the two major components, AWI-CM3 includes a river routing scheme. It receives from OpenIFS the difference between precipitation, evaporation and soil moisture over land (P-E-S), and uses a map of river basins to deliver the water to discharge points along the coastline. The current river routing component has no water storage and thus acts instantaneously. Separation of the routing component from the atmosphere and ocean is a design decision shared with EC-Earth, where the flexibility and ease of modification are core ideas. This design keeps open the option of swiftly replacing the basic runoff mapper with a more sophisticated hydrological model, such as mHM (Samaniego et al., 2010) or CaMa Flood (Yamazaki et al., 2011) in the future.

## 2.4 XIOS parallel IO server

The fourth model component of AWI-CM3 is a technical helper. ECMWF removes the parallel IO server developed by Meteo France for IFS when generating a new release of OpenIFS. This leaves OpenIFS with only the possibility to provide sequentially written GRIB file output.

While this sequential IO scheme has been used, for example, by Döscher et al. (2021), it can often reach data-throughput limits in practical applications. Furthermore, while GRIB files are common in the NWP community, the climate modelling community often uses NetCDF files. Thus, the sequential output has to be converted for most analysis tools after each simulation.

With increasing model resolution and improved use of MPI/OMP hybrid parallelization, the sequential IO overhead constitutes an ever-growing fraction of the computational cost of running OpenIFS. Recently Yepes-Arbós et al. (2021) implemented the parallel XML Input/Output Server (XIOS) 2.5 into OpenIFS for this reason, and we make use of it for AWI-CM3 to reduce the computational cost and increase the integration speed.

While XIOS takes the file writing out of the critical path of the simulation, the overhead cost of XIOS is non-zero. The main
reason is that XIOS works only in grid point space, and therefore requires spectral fields to be transformed inside OpenIFS before they can be sent to XIOS for writing. Nevertheless it provides a significant reduction of computing cost, as shown in table 1. Furthermore, XIOS enables online data postprocessing, such as vertical and horizontal interpolation, as well as temporal operators for maximum, minimum, mean etc. In doing so, XIOS reduces the number of times that files have to be written and read from disk, saving storage space, and reducing the number of job steps in the work flow. Finally XIOS allows
for output directly in NetCDF, facilitating the use of AWI-CM3 model output.

FESOM2 contains its own bespoke parallel IO routines and the integration of an external dedicated IO Library is currently not envisioned.

## 3  Coupled Model Description

The coupled climate model is constructed by combining and building on the approaches of Hazeleger et al. (2010) and
Sidorenko et al. (2015). The OpenIFS version 43R3 is common between a number of AOGCMs and ESMs with regular ocean grids currently under development, including EC-Earth4 of the EC-Earth consortium and GEOMAR's FOCI-OpenIFS (Kjellsson et al., 2020). Indeed the basic functionality of the coupling interface, as well as the future ESM component integration will be shared between EC-Earth4 and AWI-CM3. Nevertheless, some differences in the coupling strategy exist, and the setup developed for AWI-CM3 shall be detailed further here.
The ESM-Tools (version 3) infrastructure software (Barbi et al., 2021) was used to manage the configuration, compiling, and runtime-scripts of the coupled model, as well as to ensure simulation reproducibility.

### 3.1  Coupling strategy

The surface heat, mass and momentum fluxes are calculated within OpenIFS and supplied to FESOM2. Here the state variables for ocean and sea ice surface are updated accordingly. The runoff-mapper calculates its river routing after the atmosphere
component computes and provides the P - E over land for a given coupling time step.

We employ so called concurrent coupling, with surface condition updates that are considered to be numerically independent between ocean and atmosphere. The temporal exchange of the ocean and atmosphere surface conditions is taking place at the least common multiple of the ocean and atmosphere timesteps. For the TCo159L91-CORE2 simulations the timesteps for coupling, the atmospheric model and the oceanic model are 120, 60 and 40 minutes respectively, while for the TCo319L137-DART
simulation they are 60, 15 and 4 minutes. In the production mode both the atmosphere and ocean components compute their own surface update at time $t_n$ based on time lagged information at $t_{n-1}$ of the other component. The physical inconsistencies

resulting from this double-sided-lag method are small compared to those stemming from e.g. spatial and temporal truncation and are generally accepted in the climate modelling community, as they allow for parallel execution of model components (Lemarié et al., 2015; Marti et al., 2021).

AWI-CM3 can also be run in a sequential atmosphere-first mode, updating the ocean at timestep $t_n$ with atmospheric fluxes from $t_n$. In this mode climate models can get very close to what would be a converged solution of an iterative coupling at the atmosphere-ocean interface (Marti et al., 2021). Integration of a Schwarz iterative method for fully converged surface coupling is not planned, due to the high computational cost compared to small reduction in model error.

On the technical side all three components of the coupled model are compiled into their own respective executables and a parallel communication library, OASIS3-MCT 4.0 (Craig et al., 2017), is integrated into each one. The realized AOGCM setup is sketched in Figure 1. Since FESOM2 has previously been coupled to the atmospheric model ECHAM6 (Sidorenko et al., 2019), an interface for the data exchange already existed, and the grouping of fluxes in OpenIFS has been modeled after the grouping in ECHAM6. OpenIFS CY43R3 had not been coupled via OASIS before, but an older related model OpenIFS CY40R1 was coupled via OASIS as a test case in EC-Earth 3. The coupling of interface for OpenIFS CY43R3 is inspired by this predecessor. Future releases of OpenIFS will be published with this coupling interface already included.

## 4    Climatological Performance

In this section we will outline the ability of the new coupled system to reach a stable equilibrium with constant greenhouse gas and solar forcing from the year 1850. Thereafter we test to what degree the model can simulate the climate as observed over the period 1850 to 2014, with a particular focus on the last 25 years, when the observational coverage is most dense and reliable. Finally this is followed by a characterization of the response to two idealized future $CO_2$ emission scenarios, one with a sudden 4x increase of $CO_2$, and the other with a constant increase of 1% per year, starting from 1850 values.

### 4.1    Spinup drift (SPIN)

A 700-year long spinup of AWICM3 was carried out under constant greenhouse gas and solar forcing from the year 1850, starting from winter Polar Science Center Hydrographic Climatology (PHC3) (Steele et al., 2001). The forcing fields were collected from the input4MIPs data server (https://esgf-node.llnl.gov/search/input4mips/, last access: 6 November 2019). Aerosol fields were kept at present day levels, as the integration of the emissions-based aerosols into OpenIFS CY43R3 through the EC-Earth Consortium with tracing via the M7 model (Vignati et al., 2004) is still ongoing. This implies a somewhat colder pre-industrial state in particular in regions of the northern hemisphere, that are cooled in present day observations due to industrial aerosol emissions.

During the first 500 years of the spinup simulation we noted positive global ocean temperature trends throughout nearly the entire water column, as can be seen in the Hovmöller diagram of Figure 3b. Evaluation of the top of atmosphere (TOA) and surface (SFC) net heat fluxes in the atmospheric model revealed a near constant radiative imbalance of $2 W/m^2$, depicted in Figure 3a. A partial solution is to switch the OpenIFS mass fixer from dry mass to total mass conservation (Malardel et al.,

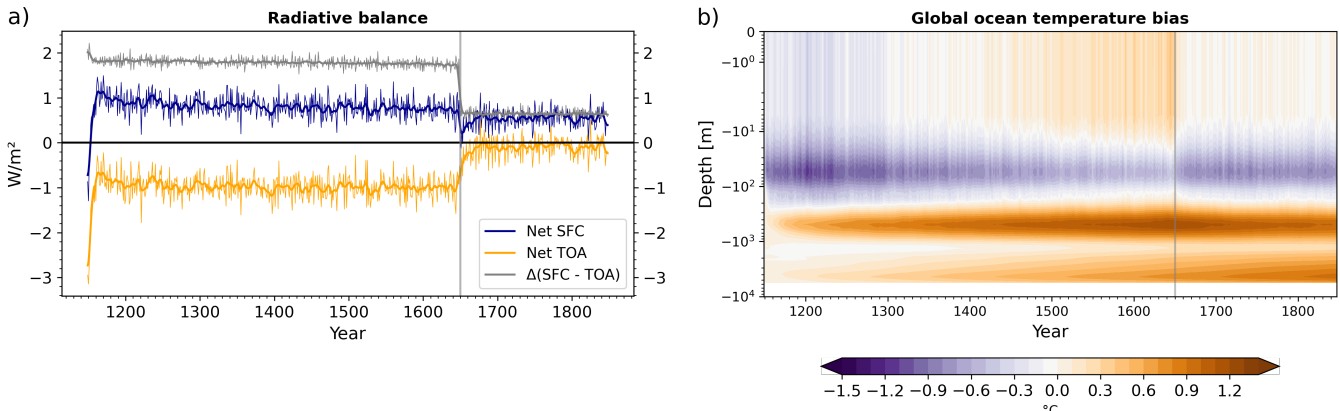

**Figure 3. a)** Net radiative imbalance at the top of atmosphere (TOA) and at the surface (SFC) in the spinup simulation. Positive (negative) values indicate downwards (upwards) net heat flux. The difference results from spurious energy production in the atmosphere. **b)** Semi-logarithmic depth Hovmöller diagram of the evolution of the global mean ocean temperature bias over the spinup period with respect to PHC3 climatological values (Steele et al., 2001). Switching from dry to total mass fixer after 500 years reduced spurious heat production in the atmosphere and halted the ocean warming trend in the upper 1000 meters. Warming at depth continued.

2019) with the McGregor scheme (McGregor, 2005). We implemented this solution starting from year 1651. Subsequently the
radiative imbalance reduced to $+0.7\,W/m^2$, which is within the range of imbalance of CMIP6 models (Wild, 2020).

Further experiments showed that decreasing the OpenIFS timestep from 60 minutes to 30 minutes reduced the imbalance to $+0.4\,W/m^2$. An additional reduction to 15 minutes timesteps showed no more improvements. We surmise that the timestep-dependent component of the error implicates the semi-lagrangian trajectory algorithm operating close to the stability limit for the given atmospheric resolution of TCo159. For our analysis we judged this additional error an acceptable price for a doubling
in model integration speed, and we thus keep the 60 min timestep. The timestep independent flux imbalance of $+0.4\,W/m^2$ will be targeted in future model development and tuning efforts.

In the global mean ocean temperature Hovmöller diagram (Figure 3) we can see that after switching to total mass conservation the reduction of spurious heat production in the atmosphere led to a stabilization of the ocean temperatures in the upper 1000 meters. The trend in the deep ocean, strongest at 4500 meters depth, on the other hand has hardly slowed down and, thus,
likely has a different origin. One candidate currently under investigation is the topography-influenced equilibrium depth of the Gibraltar Straight overflow. Alternatively we speculate that overestimated mixing from the KPP mixing scheme (Large et al., 1994) might be the reason. The bias pattern is very similar to that of the previous FESOM2-ECHAM6 (AWI-CM2) coupled model (Sidorenko et al., 2019).

Another potential contributor to the accumulation of heat at depth is a consistent positive shortwave radiation bias of
OpenIFS in the Southern Hemisphere. Some of the spuriously heated Southern Ocean surface water gets entrained into the Antarctic Intermediate Water (AAIW). In recent years, the Southern Ocean shortwave radiation bias has been the subject of research which led to its reduction by $5-10\,W/m^2$ (Forbes et al., 2016). We have already back-ported these improvements

originally developed for the operational IFS CY45 model into OpenIFS CY43R3 prior to our spinup simulation. Even with the improvements, a positive shortwave downward radiation bias of up to $5 - 10 \, W/m^2$ between 45-60°S remains, and can be

seen in Figure 7 e). Idealized experiments have shown that removal of the remaining shortwave radiation bias would cool the Southern Ocean by roughly $1 \, K$ within a decade, with potentially larger improvements on longer timescales (not shown).

## 4.2 Pre-industrial control (PICT)

As evident from the Hovmöller diagram (Figure 3b), the spinup run is not yet in equilibrium at depths greater than 1000 meters. A small residual drift can also be found at the ocean surface and in the atmosphere, which can be seen as a consequence

of the still-drifting deep ocean. Based on experience with other ocean models we can estimate that a 3000-5000 year long simulation would be needed for the model to reach full equilibrium (Rackow et al., 2018). Instead, we run a pre-industrial control experiment which serves as a reference for correcting the historical-period simulation with respect to the remaining trends. The pre-industrial control run thus extends the spinup run by 165 years with the same year 1850 greenhouse gas and solar forcing. We construct a simple linear regression model for the remaining drifts in PICT and subtract these when we

analyze the response to historical forcing in the historical simulation.

## 4.3 Climatological performance during the historical period (HIST)

In the following we first characterize the most prominent bias patterns of the last 25 years of the historical simulation with regards to reanalysis and satellite data products. We then take a look at the climate response to the historical forcing in comparison to the pre-industrial control.

For a succinct overview of the model performance we calculated climate model performance indices, based on Reichler and Kim (2008) and shown in Figure 4. For the four seasons, seven regions and twelve key variables, the fraction of absolute error of climatology of the last 25 years in AWI-CM3 historic simulation to the absolute error averaged over 30 CMIP6 contributing models can be seen. A complete list of the CMIP6 models serving as the evaluation set is given in Appendix A. The list of observational datasets used to calculate all mean absolute errors is also given in Appendix A. For all model

and observational datasets, where available, the time period from December 1989 until November 2014 is considered. As the performance of AWI-CM3 is expressed as a fraction of the errors of the CMIP6 average performance, values below 1 indicate better performance, values above 1 point to larger model errors.

For the majority of variables, seasons and regions our post-CMIP6 prototype model is already performing better than the average CMIP6 model. The lead is especially large for cloud cover (clt) and 500 hPa geopotential height (zg). For surface

meridional wind (vas), surface zonal winds (uas), 300 hPa zonal wind (ua), TOA outgoing longwave radiation (rlut), and precipitation (pr) the bias to observations is mostly below average. The very important near surface (2m) air temperature (tas) is simulated well in the Arctic (60°-90°N) and reasonably well in the northern mid-latitudes (30°-60°N) and tropics (30°S-30°N). In the southern mid-latitudes (30°-60°S) and Antarctic (60°-90°S) the air temperature bias is relatively high. A look at the sea ice concentration (siconc) reveals that the respective errors are far above average in the Antarctic. The problem is particularly

severe in the austral winter season. This bias is in co-occurrence with a large mixed layer depth bias in the Antarctic. We note

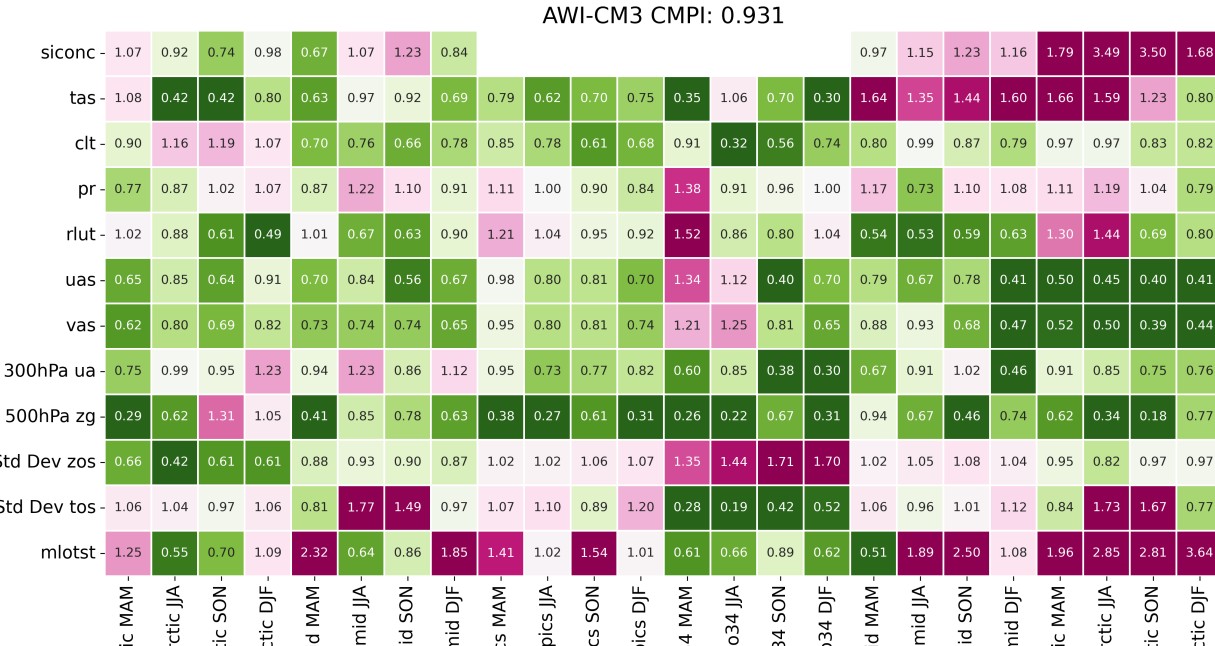

**Figure 4.** Performance indices after Reichler and Kim (2008) that give the fraction of absolute error of climatology of the last 25 years in AWI-CM3 historic simulation to the absolute error averaged over CMIP6 models. Values below (above) one correspond to below (above) CMIP6 average biases. The underlying observations against which all models were evaluated are OSISAF OSI-450 (Lavergne et al., 2019): sea ice concentration (siconc); MODIS Atmosphere L2 Cloud Product (Platnick et al., 2015): cloud cover (clt); Global Precipitation Climatology Project (GPCP) Monthly Analysis (Adler et al., 2018): precipitation (pr); Clouds and the Earth's Radiant Energy System (CERES) (Wielicki et al., 1996): TOA outgoing longwave radiation (rlut); ECMWF re-analysis ERA5 Hersbach et al. (2020): near-surface air temperature (tas), eastward near-surface wind (uas), northward near-surface wind (vas), 300 hPa eastward wind (ua), 500 hPa geopotential height; NOAA JASON-1, JASON-2, CryoSat-2 combined: Sea surface height (zos); HadISST2 (Titchner and Rayner, 2014): Sea surface temperature (tos); C-GLORSv7 (Storto et al., 2016): Mixed layer depth (mlotst). A full list of the considered CMIP6 models is given in Appendix A.

that the mixed layer depth from reanalysis we used here as the reference could be associated with uncertainties too, as it is based on the implementation of mixing parameterizations in the reanalysis model. The standard deviation of sea surface height (zos) shows reasonable variability of the ocean currents in the mid and high latitudes, with problems found in the Nino34 region (5°S-5°N, 170°W-120°W). On the other hand, the standard deviation of temperature is best represented here, with deficiencies in the Antarctic cold season and northern mid latitude summer.

A simple average over all individual performance indices gives AWI-CM3 a score of 0.931, with 15 out of the 30 considered CMIP6 models performing better. The overall index of the AWI-CM3 prototype simulation is improved compared

to its CMIP6 predecessor with similar resolution and computational cost (AWI-ESM1.1-LR, 1.044). The performance of our medium-resolution AWI-CM1.1-MR CMIP6 contribution (Semmler et al., 2020) is better (0.894), but this model configuration is 20 times more expensive to run than the simulations presented here. Preliminary tests with higher resolution at equal computational cost indicate that AWI-CM3 can achieve better climatological performance than AWI-CM1.1-MR, as we will show in Section 5.1. A major contributing factor are the faster dynamic cores and better computational scalability of the model components, allowing for higher resolutions at equal computational cost.

With this overview in mind we will limit our model bias analysis to problematic areas, knowing that for the variables we do not focus on we achieved good model performance.

### 4.3.1 Sea ice and mixed layer depth

The sea ice thickness in the Arctic contains realistic values for both end of summer (EOS) and winter (EOW), PICT and HIST runs (Figure 5). In the central Arctic, PICT sea ice thickness ranges from 3.5m at the EOW to 2.5m at the EOS. The mean over the last 25 years of the HIST simulation reveals a reduction of sea ice thicknesses to 2.5m and 1.5m in these two seasons. In both simulations, the maxima of EOW ice thickness can be found in the East Siberian Sea and along the coastline of northern Greenland. Although, in some years, a local maximum of sea ice thickness was observed along the East Siberian Sea coast, in the multi-year-mean field it should not be as large as presented by the model. Similar issues are common in many CMIP6 models and exist even in the PIOMAS reanalysis (Watts et al., 2021). One clear bias in comparison to observations is a too wide tongue of sea ice extending eastward in the Greenland Sea during the winter months. A similar feature can be seen in our previous model versions. The annual cycle of sea ice extent is represented well with 16 to 7 million square kilometers in the last 25 years, compared to observational values of 15 to 6 million square kilometers by Walsh et al. (2019).

The Antarctic sea ice biases require the most improvement as follows from the metrics presented in Figure 4. In both the PICT and HIST simulations the sea ice covered area is strongly underestimated during austral winter (Figure 5c&g). Notable are two spots of low sea ice thickness in the Weddell Sea and in the eastern Ross Sea. Both areas feature low EOW mean sea ice thickness of less than 20cm during the PICT run, and are partially ice free during the last 25 years of HIST. These locations feature persistent large scale polynyas. In reality, polynyas were observed in the Weddell Sea (e.g., during the winters 1974 to 1976), however, the frequency of their occurrence (not presented in this paper) is clearly overestimated in our model.

In order to gain an understanding towards the reasons for the persisting polynya, we investigate the mean mixed layer depth (MLD), defined as the depth at which the potential density differs by 0.125 $\frac{kg}{m^3}$ from the surface density (Monterey and Levitus, 1997). The MLD shown in Figure 6 a) features large values in the Weddell and increased values in the Ross seas that are co-located with low sea ice concentration values seen in figure 6 b). We therefore speculate that the large MLD could be one of the reasons for the underestimated wintertime sea ice, as the ocean heat from the warmer Circumpolar Deep Water (CDW) can be mixed up to reach sea ice from below. The salinity profile over the highlighted region confirms that the surface layer is more saline in our model than the PHC3 climatology. The exact reason for the overestimated MLD is not clear and understanding whether the salinity bias is the main cause is among our planned future research.

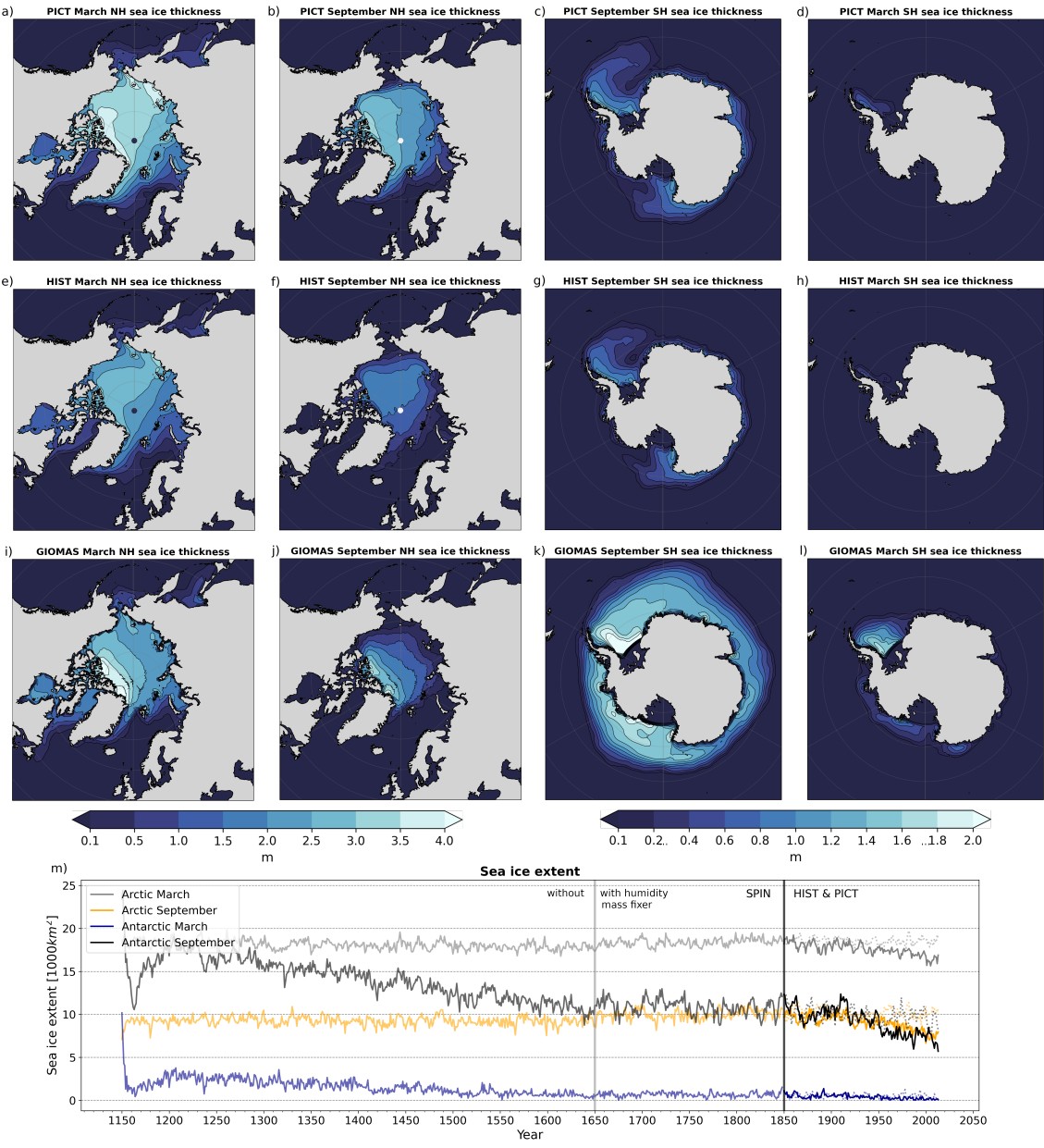

**Figure 5. a-d)** Mean sea ice volume per unit area over last 25 years of PICT simulation. **e-h)**: Same as top row but for the HIST simulation. **i-l)**: Same as top row but for GIOMAS sea ice thickness reanalysis product. (Zhang and Rothrock, 2003). **m)**: Evolution of sea ice extent over SPIN, as well as HIST and PICT simulations. Mass fixer switched from dry to total in years 1650. HIST and PICT forcing applied from 1850 onwards.

We hypothesize that in the subsequent austral summer the reduced sea ice cover results in a strong positive sea ice albedo feedback, further heating up the surface ocean. We examine this feedback in the following section.

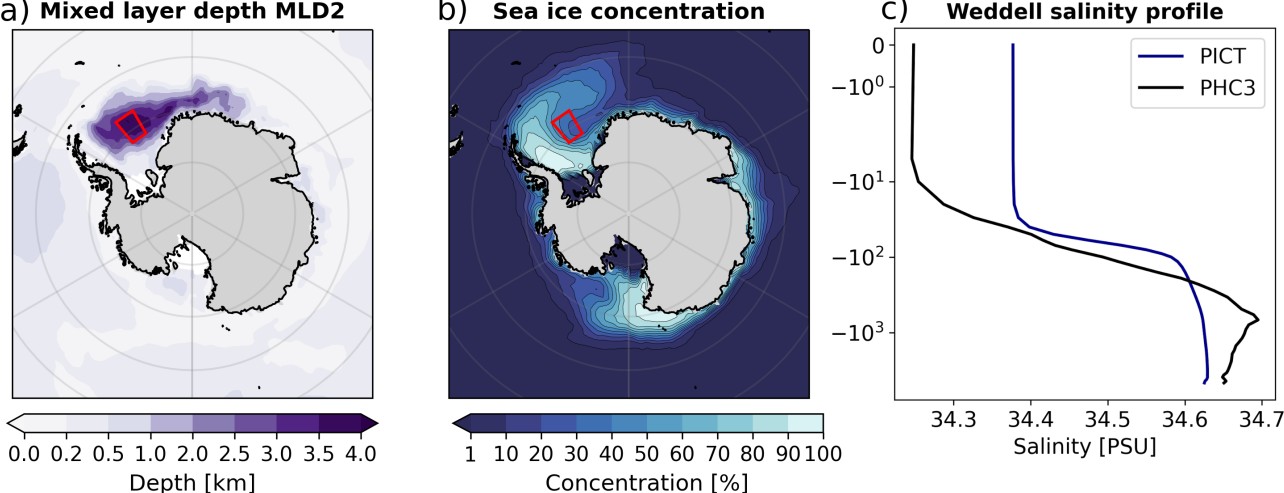

**Figure 6. a)** Austral winter mean mixed layer depth (MLD) where the potential density over depth differs by 0.125 $\frac{kg}{m^3}$ from the surface density (Monterey and Levitus, 1997) averaged over the last 25 years of the PICT simulation. **b)** High MLD values in the Weddell and Ross seas contribute to the persistent polynyas visible in the mean austral winter sea ice concentration over the same time period. **c)** Salinity profile over the region marked in red in comparison to PHC3 climatology (Steele et al., 2001).

### 4.3.2 Surface temperature and fluxes

The near surface (2m) air temperature bias in comparison to ERA5 largely shows a pattern similar in sign (Figure 7a), and at somewhat higher amplitude compared to the CMIP6 multi model mean bias shown in Bock et al. (2020). In the tropical Pacific we find a slight cold bias located at the equator, flanked by equally sized warm biases, which is typical of a too-strong double-ITCZ found in many coupled models and associated with characteristic percipitation and short-wave radiation biases (see below). The upwelling regions off the west coasts of southern Africa and South America feature well known warm biases of up to 4°C resulting from insufficient upwelling, too little stratocumulus cloud cover and thus too much downwelling short-waver radiation with our low resolution ocean model. Further north and south the subtropical gyres feature cold biases on the order of -1 to -2°C.

In the mid-latitudes, the most prominent bias is the misplacement of the Gulf Stream which fails to represent the correct North West Corner detachment (Figure 7a). This is a well known bias in OGCMs of coarser than 10 km resolution in the North Atlantic. For the FESOM2 model, a study by Sein et al. (2017) shows that this problem can be mitigated by increasing the ocean model resolution in the region.

At high latitudes we find a strong warm bias of more than +8°C over areas of the Southern Ocean (SO) adjacent to Antarctica and a moderate one over Antarctica of +3 to +4°C, as well as a cold bias of around -2°C in the Arctic. The cold bias in the Arctic likely stems from the ERA5 reanalysis, which misses snow cover on sea ice (Batrak and Müller, 2019), rather than from AWI-CM3. We thus focus on the southern warm biases in our analysis, some of which are well known for IFS based climate

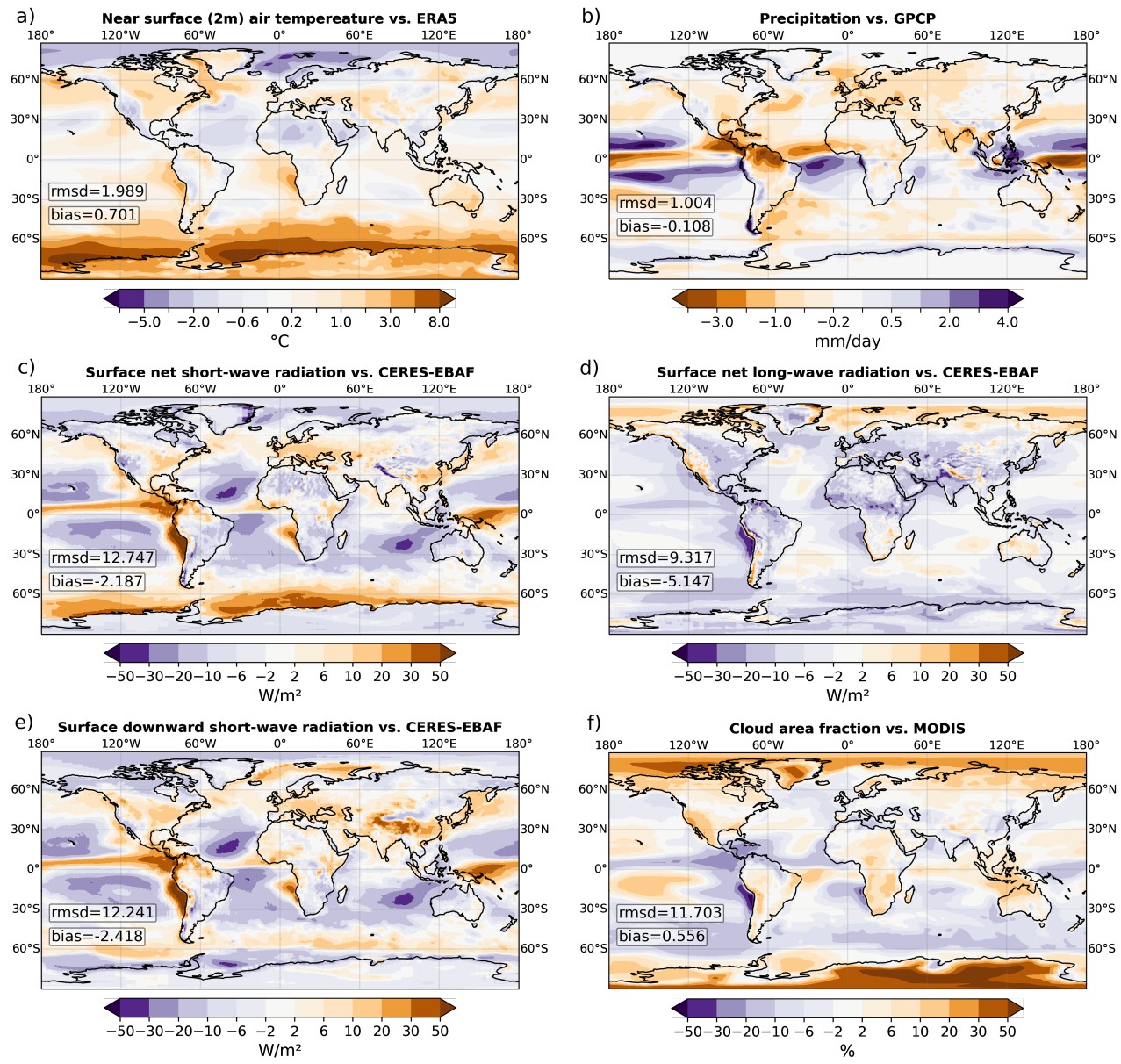

**Figure 7.** Annual mean **a)** Near surface (2m) air temperature bias with respect to ERA5. **b)** Precipitation bias with respect to GPCP. **c)** Surface net shortwave radiation bias with respect to CERES (2001-2014). **d)** Surface net longwave radiation bias with respect to CERES (2001-2014). **e)** Surface downward shortwave radiation bias with respect to CERES (2001-2014). **f)** Total cloud cover fraction bias with respect to MODIS. For each variable rmsd gives the area weighted root mean square distance between observations and model data, and bias gives the area weighted mean distance.

models and can also be seen in Döscher et al. (2021) and Roberts et al. (2018). We hypothesize that these biases are caused by the direct effect of heat released from a spuriously deep mixed layer (as noted in Section 4.3.1), a positive ice albedo feedback to the resulting reduced sea ice cover, and a remaining positive net shortwave downward heat flux bias between 45-60°S.

Nearly all of the near surface temperature biases (Figure 7a) can be associated with co-located net surface shortwave radiation biases (Figure 7c). These can be largely explained by surface downward radiation biases (Figure 7e), which in turn result from total cloud cover fraction biases (Figure 7f). This linkage pattern is well known in the modelling community (Satoh et al., 2019) and will likely remain a dominant source for surface temperature biases until deep convection resolving atmospheric models become readily available for multidecadal to centennial climate simulations.

There are two notable exceptions to this causal chain in our simulations. Firstly, we found a cold bias over the Greenland sea where the surface downward shortwave radiation bias is positive. The cold bias in the region is the result of a lobe of sea ice drifting into the area from the Greenland coast during winter. This results in an overestimation of surface albedo and a reduction of the net surface shortwave radiation.

The second and similar exception is the previously mentioned warm bias south of 60°S in the SO. While the model over-estimates the cloud and thus underestimates surface downward shortwave radiation in the region, the sea ice fraction in this area is too low. The surface net shortwave radiation and the near surface temperature biases are therefore positive. As the SO surface net shortwave radiation bias is negative the low sea ice concentrations can not originally be caused by shortwave biases. Indeed, the near-surface (2m) air temperature and sea ice concentration biases south of 60°S peak during the austral winter season, while the biases in the southern mid-latitudes peak during austral summer (Figure 4). Correcting the largest biases south of 60°S in our model will probably necessitate work on non-solar heat fluxes and mixed layer depths at high-latitudes.

The precipitation biases shown in Figure 7b feature the canonical double ITCZ bias in the Tropics. Notably the mid-latitudes receive too little precipitation, especially over western boundary currents, Europe and North America east of the Rocky Mountains.

Figure 8 a) depicts the zonal mean temperature bias averaged over the last 25 years of the HIST simulation with respect to ERA5 for the same period. The previously seen Southern Ocean surface warm bias is well visible up to 500 hPa. Around the tropopause height the model exhibits a prominent cold bias of up to -4°C. In the high stratosphere meanwhile we find a large warm bias of up to 8°C. The stratospheric bias will be reduced with ECMWF's next minor release of OpenIFS, cy43r3v2, which enables the use of spectral solar insolation instead of total solar irradiance (not shown).

The zonal mean zonal wind in figure 8 b) broadly shows overestimation of the subtropical jet strength and height, as well as underestimation of the polar jet streams. The most notable feature though is the miss-representation of equatorial stratospheric winds.

A closer look at the near equator (10°N-10°S) zonal and meridional mean wind-speeds over height and time in figure 8 c) reveals that the Quasi-Biennial Oscillation (QBO) is in fact an annual oscillation in the last 25 years of the HIST simulation. Contrasted with the same winds in the ERA5 reanalysis of the equivalent time period the QBO frequency error is apparent. The main reason for this behaviour is a lack of tuning for the gravity wave flux parameterization, as was confirmed by tests

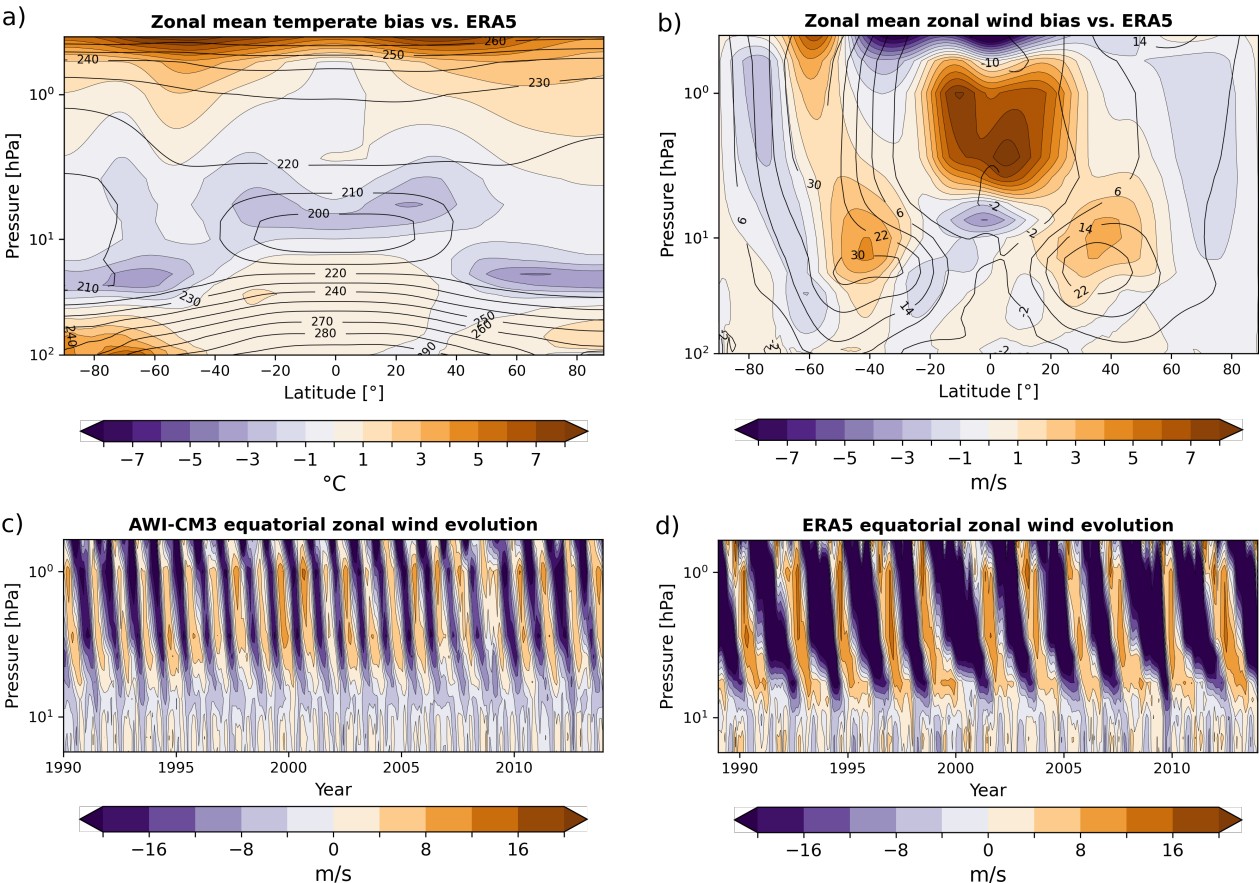

**Figure 8.** AWI-CM3 HIST **a)** Zonal mean air temperature bias with respect to ERA5 averaged over the last 25 years. **b)** Zonal mean zonal wind bias with respect to ERA5 averaged over the last 25 years. **c)** Near equator zonal wind over time and height for AWI-CM3 HIST simulation. **d)** Same as c) but for ERA5 reanalysis, showing Quasi-Biennial Oscillation.

conducted with OpenIFS at GEOMAR (not shown). Tuning of the gravitational wave flux parameterization will be addressed
in future model releases.

### 4.3.3 Ocean temperature

On the ocean side of the coupling interface, we find mean sea surface temperature biases with respect to PHC3 that are nearly
identical to the aforementioned near surface air temperature biases (Figure 9). At a depth of 100 meters the bias looks similar
to that at the surface in high-latitudes where mixed layer depths are large. In the Tropics and Subtropics this depth range is
dominated by cold biases, which could be partially due to missing vertical mixing associated with Langmuir circulation in the
current version of FESOM2. It is known that including a parameterization for this mixing can effectively alleviate the cold bias

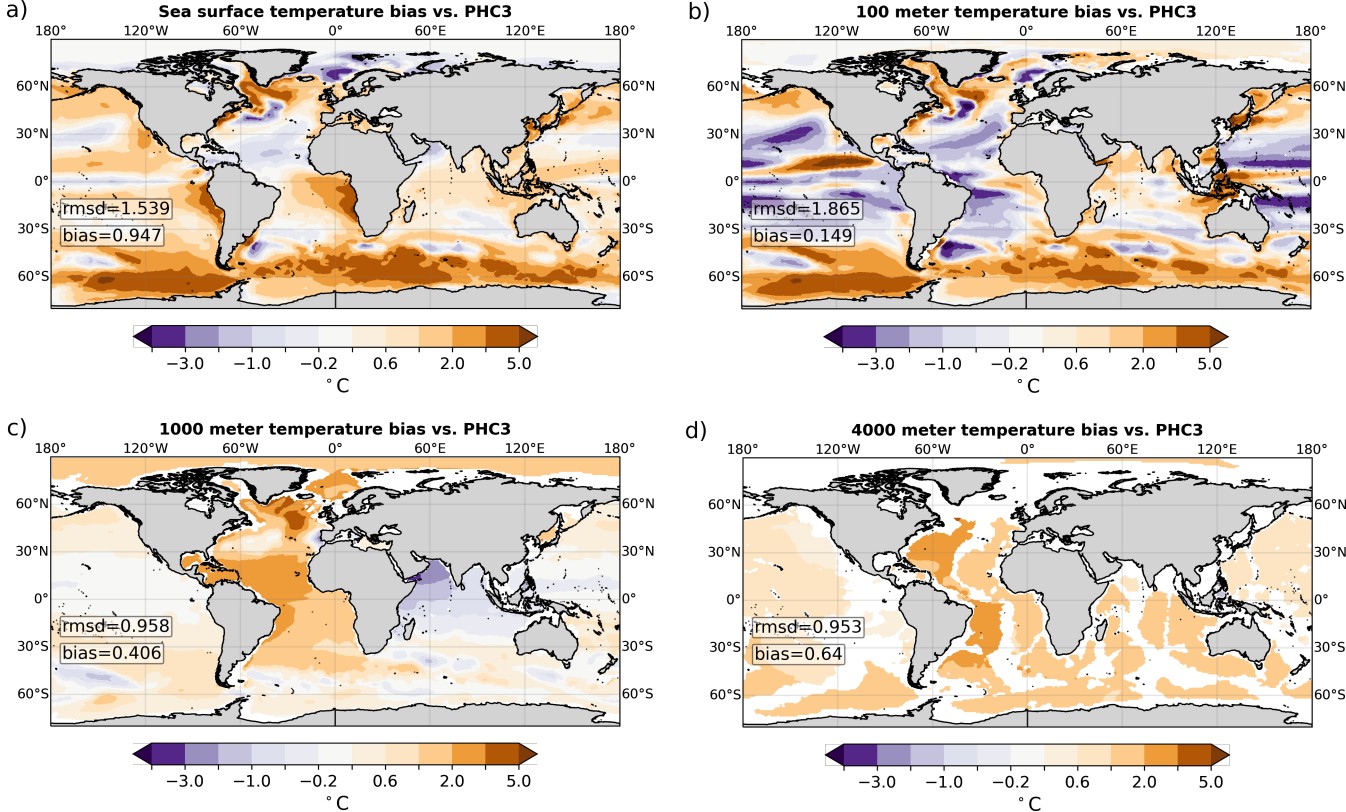

**Figure 9.** Mean ocean temperature bias of the last 25 years of the historical simulation with respect to PHC3 at **a)** 0, **b)** 100, **c)** 1000 and **d)** 4000 meters depth. For each depth rmsd gives the area weighted root mean square distance between observations and model data, and bias gives the area weighted mean distance. At the surface the ocean-resolution-specific warm biases in upwelling regions and western boundary currents can be seen, in addition to a pronounced Southern Ocean warm bias. At depth, a warm bias is dominating in the Atlantic.

in the near-surface ocean in the mid-latitudes and tropics (Wang et al., 2019; Ali et al., 2019), and the integration of a Langmuir circulation parameterization in vertical mixing schemes is planned.

At a depth of 1000 meters the Atlantic shows a strong warm bias, which we speculate results partially from spuriously warm SO surface water entrained into the AAIW. Ultimately the whole bias probably stems from multiple yet to be identified sources in both hemispheres. The cold bias found in the Indian Ocean at this depth is likely the result of insufficient warm outflow from the Persian Gulf because the narrow strait is not well resolved on the FESOM2 CORE2 mesh.

AWI-CM3 shows a pronounced warm bias of 3-5K in the Atlantic subpolar gyre. Sidorenko et al. (2015) noted a similar bias when analyzing AWI-CM1 simulations. This bias is shared between many climate models which contributed to CMIP. Hence, a similar drift in ocean hydrography is also described in Sterl et al. (2012); Delworth et al. (2006, 2012); Jungclaus et al. (2013). These authors discuss different factors that may be responsible for the bias. Sterl et al. (2012) show that overestimation of the Mediterranean outflow can significantly increase the deep-ocean salinity bias. Delworth et al. (2012) attribute this anomaly to

the insufficient eddy transport required to compensate for the wind-driven subduction in the subtropical gyres. They show that moving towards an eddy-resolving setting or a parameterization of the eddy stirring reduces the temperature biases significantly. Jungclaus et al. (2013) suggest that part of the problem arises from the improper interbasin exchange between the Indian and South Atlantic oceans.

At 4000 meters depth we observe the warm biased Atlantic water mass spreading into the whole global ocean. As the meridional circulation at this depth is northward in the Antarctic Bottom Water (AABW) cell, the origin of the warm bias is presumably insufficient cold AABW formation in the Southern Ocean on this coarse mesh. The slowness of the AABW circulation in the Atlantic coincides with the slow but continued global-mean warming trends at great depths seen in Figure 3. Biases presented in Figure 9 explain the Hovmöller diagram in Fig 3 which we addressed in Section 4.1. The three sets of anomalies in the Hovmöller diagram at depths 100m, 1000m and 4000m stem from the biases in the mid latitudes, the North Atlantic and in the entire ocean, respectively.

### 4.3.4 El Niño-Southern Oscillation

To assess the model capabilities in representing climate variability, we investigate the representation of the El Nino Southern Oscillation (ENSO). Using the entire HIST simulation and the EOFs toolsbox (Dawson, 2016) Figure 10a) depicts the spatial extent of the correlation between the first principle component and the input data set at each point in space. Also depicted is the Nino3.4 box (5°N to 5°S, 120°W to 170°W) upon which we based our further analysis.

We calculated the area mean SST within the box, applied a linear detrending, subtracted the mean seasonal cycle, and finally computed a three month running mean. The resulting Nino3.4 index timeseries is shown in Figure 10c. Comparison with the Nino3.4 index based on the HadISST observational estimates (Figure 10d) reveals that our simulated amplitude of ENSO, with a range of +1.9 to -1.6°C, is lower than the observed. Indeed, the histograms in Figures 10e and 10f reveal missing tails of the distribution in the simulation.

While the overall strength of ENSO is underestimated by AWI-CM3, the normalized power spectrum density (PSD) in Figure 10b shows that the frequencies of the observed HadISST ENSO are well reproduced. We find several peaks concentrated between periods of 2.8 and 12 years. Further analysis (not shown) indicated that ENSO phase locking is rather weak in the model version presented, and that both ENSO amplitude and phase locking can be improved through reduction of equatorial Pacific precipitation and temperature biases. Understanding how to do so without negatively impacting model performance in other regions is work in progress.

Future improvements for the ENSO amplitude could also potentially be achieved by activating the OpenIFS internal stochastic parameterization schemes, as shown by Yang et al. (2019). Indeed, HighResMIP simulations performed by Roberts et al. (2018) with IFS CY43 that employed the Stochastically Perturbated Parametrisation Tendencies (SPPT) scheme (Buizza et al., 1999) had an ENSO amplitude more in line with observational estimates. Since these studies used the SPPT scheme, they required additional humidity mass fixers for climate-length integrations.

Alternatively, in order to use the new - and from a mass conservation perspective more favorable - Stochastically Perturbed Parametrisations (SPP) scheme (Ollinaho et al., 2017), the full version of SPP has to either be backported from IFS CY47 to

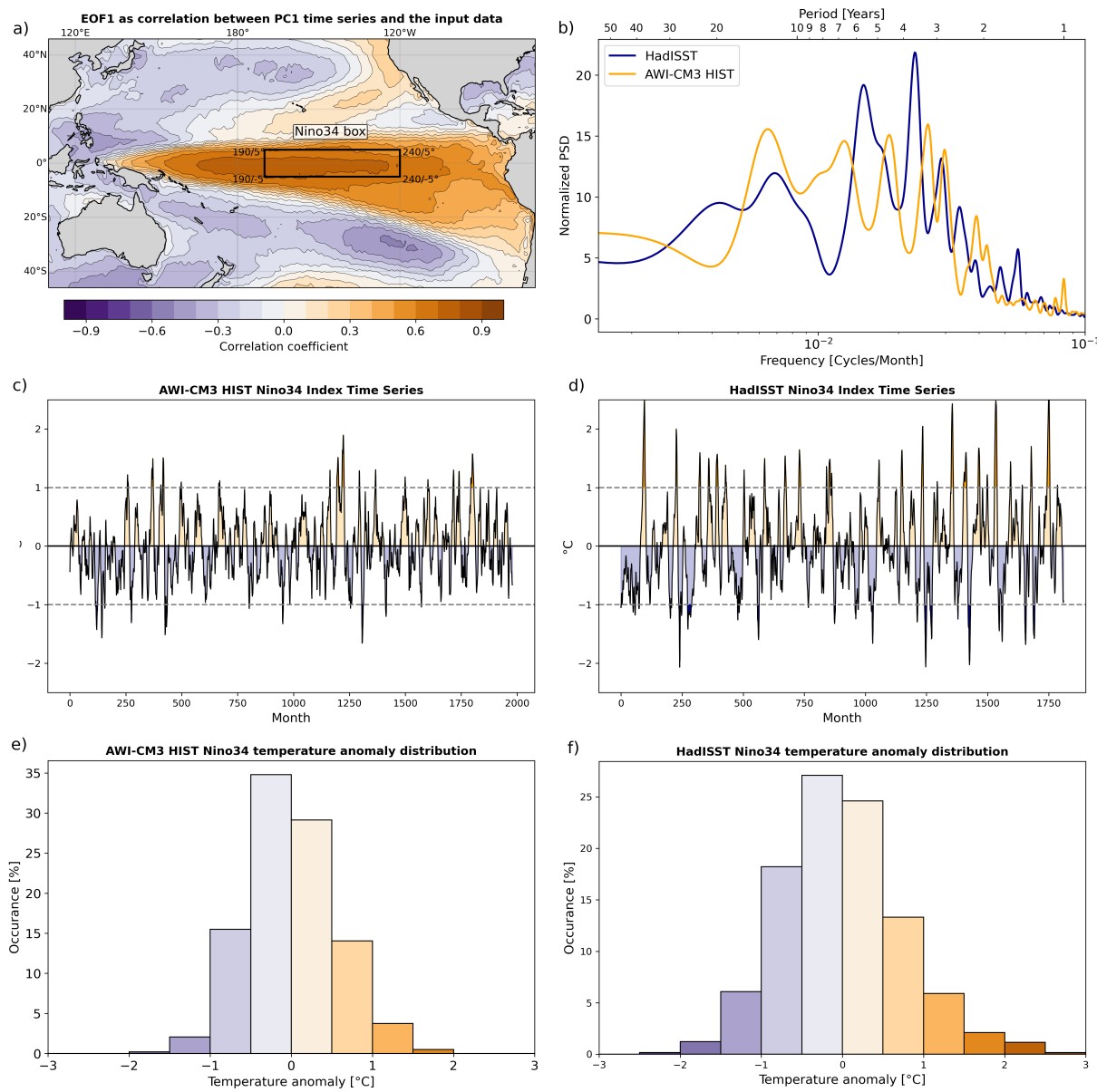

**Figure 10. a)** Extent of the Nino3.4 box, as well as the first empirical orthogonal function expressed as the correlation between the first principle component and the input data set at each point in space. Calculated from the entire HIST simulation using the EOFs toolsbox (Dawson, 2016). **b)** Normalized power spectrum density of simulated ENSO within the Nino3.4 box in comparison to HadISST observational estimates for the years 1850 to 2014. **c)** & **d)** Nino3.4 SST index defined as the detrended, seasonal cycle removed and 3-months running mean timeseries of SST averaged over the Nino3.4 box. The results from AWI-CM3 HIST and HadISST observational estimates are shown, respectively. **e)** & **f)** As c) & d) but depicting occurrence of temperature anomalies in bins with a width of $0.5°C$.

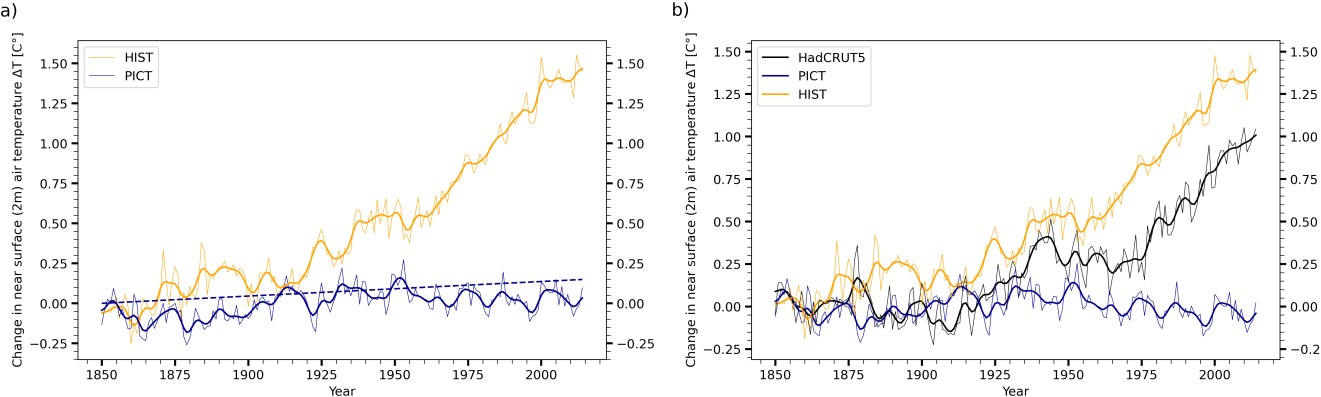

**Figure 11. a)** Near surface (2m) air temperature changes over the historic period and in the pre-industrial control simulation. Thick lines show the 10 year running means. The pre-industrial control simulation shows a residual drift of $0.00091 \frac{°C}{\text{Year}}$ obtained from linear regression. **b)** Both simulations have been corrected by subtracting the residual drift. The resulting simulated global warming over the period 1850 to 2014 is $1.4°C$. In comparison to observational estimates from HadCRUT5 (Morice et al., 2021) warming is overestimated by $0.4\,\text{K}$, likely due to fixed aerosols in the AWI-CM3 prototype.

OpenIFS CY43R3, or the new version OpenIFS CY47 has to be released. Noteworthy is also that CMIP6 simulations done with CNRM-CM6-1, featuring the IFS atmosphere with the ARPEGE physics package (not in use AWI-CM3, as we employ ECMWF physics) and no stochastic parameterizations, also had a high ENSO amplitude. However the frequency of ENSO was too focused with a sharp single peak in the 3-4 year band (Voldoire et al., 2019).

Finally experiments with stochastic coupling at the atmosphere ocean interface conducted with AWI-CM1 yielded improved ENSO phase locking (Rackow and Juricke, 2020), providing another potential development avenue.

### 4.4 Impact of historical forcing

#### 4.4.1 Air temperature

After we have established that AWI-CM3 behaves reasonably for much of the globe and many important climate parameters, we will now characterize the impact of historical greenhouse gas and solar forcing on the evolution of several of these variables.

The global mean near surface (2m) air temperature increases under HIST forcing, as seen in Figure 11a. Since our SPIN experiment did not establish a full equilibrium in the deep ocean, we analyze the air temperature change in our pre-industrial control experiment PICT, and obtain the residual drift of $0.00091°C/\text{Year}$ via linear regression. Over the 165-year-long simulation this amounts to a drift of $0.15°C$ in the PICT run.

In Figure 11b we have corrected both runs by deducting the linear trend. The resulting temperature change represents a global warming over the period 1850 to 2014 of $1.4°C$, most of which comes in a steep rise between 1960 and 2000. Comparison with

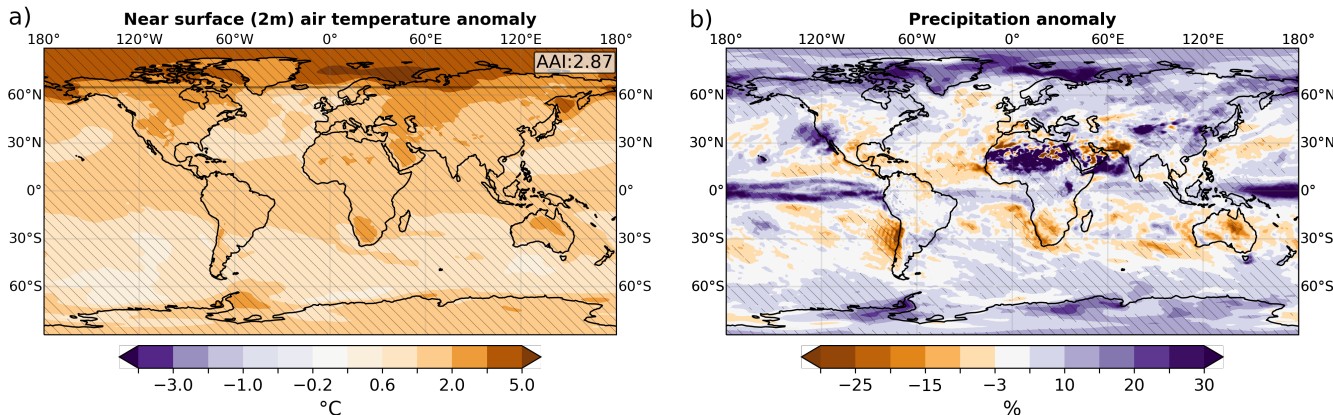

**Figure 12. a)** Temperature anomaly between last 25 years of the historic simulations and the equivalent period of the pre-industrial control run. The Arctic Amplification index, defined as the ratio of warming north of 65°N expressed as a fraction of global warming, is 2.87. Significant temperature anomalies are hatched and the 95% significance is obtained via bootstrap test. **b)** Same as a) but for relative precipitation changes.

observational values from HadCRUT5 (Morice et al., 2021) shows that AWI-CM3 gets the timing of historical warming spikes right, however the strength is overestimated by $40\%$ ($1.0°C$ vs. $1.4°C$).

The main reason for the stronger than observed historical warming is that the increase in global aerosol emissions, which partially masks the warming induced by well mixed greenhouse gases, is not incorporated in the version of OpenIFS (CY43R3) used here. The integration of the tropospheric aerosol component M7 into OpenIFS CY43R3 is still ongoing within the EC-Earth consortium (private communication with the EC-Earth aerosol working group). The sixth IPCC report (Masson-Delmotte et al., 2021) concludes that the anthropogenic aerosols direct and indirect contributions to the effective radiative forcing amount

to about $-0.5W/m^2$. Comparing to the total anthropogenic forcing of $+1.5W/m^2$, we foresee that, once transient aerosols will have been included in OpenIFS CY43R3, AWI-CM3 historical global warming will be much closer to the observed value.

The map of temperature changes in Figure 12a shows that under historic well-mixed greenhouse gas and solar forcing AWI-CM3 simulates a temperature increase of approximately $1-2°C$ in the Tropics, Antarctic and large parts of Eurasia. Mid-latitude oceans in both hemispheres see a weaker warming of $0.6-1°C$. Larger warming of $2-3°C$ can be found in the Middle East,

North America, the Caucasus, Australia and South Africa. Sea ice covered regions in the Arctic experience the strongest air temperature increases, with the Arctic (65-90°N) warming by $3-8°C$. Defining an Arctic Amplification Index (AAI) as the ratio of warming north of 65°N to whole northern hemisphere warming (Davy et al., 2018; Johannessen et al., 2016), the resulting AAI is 2.87. Note that, in contrast to observations as well as full-forcing CMIP simulations, there is no trace of a warming hole in the North Atlantic south of Greenland. Whether this might be due to the missing transient aerosol forcing in our simulation

is unclear. Earlier studies have linked the warming hole rather to a weakening of the AMOC (Keil et al., 2020), but although our historic simulation does exhibit such an AMOC weakening (see Figure 13), no warming hole forms.

### 4.4.2 Precipitation

Figure 12b shows the simulated changes in the precipitation pattern resulting from historic well-mixed greenhouse gas and solar forcing. The most important features are: the high latitudes nearly uniformly receive more precipitation; the monsoonal precipitation in North Africa and China intensifies; the ITCZ is enhanced in the western Pacific and more focused on the equator in the eastern Pacific and in the Atlantic; considerable parts of the subtropics tend to receive less precipitation. These patterns are largely consistent with precipitation changes simulated in CMIP6 models where transient aerosols are included, although precipitation increases over the Indian Ocean and Northern Central Africa are not as pronounced as in the CMIP6 model mean and more pronounced over the Indonesian warm pool (compare with Figure SPM.5c in Masson-Delmotte et al. (2021)).

### 4.4.3 Ocean circulation

The AWI-CM3 simulations reasonably reproduce the canonical pattern of the AMOC streamfunction, with an upper cell consisting of northward surface flow as well as southward return flow of North Atlantic Deep Water, and a lower cell representing the northward flow of AABW (Figures 13 a,b). During the spinup simulation the maximum of the northward transport between 30-45°N in AWI-CM3 fluctuates at around 20 Sv (see Figure 13c.). Initially the AMOC gradually slowed down to 18 Sv while the upper ocean was spuriously warming, as described in Section 4.1. After applying the total mass fixer starting after 500 years into the SPIN experiment the AMOC is recovered to 20 Sv along with the cooling of the upper ocean (Figure 3b).

Figure 13d shows a rather strong decline of the AMOC strength over the last 70 years of the HIST simulation, while the multi decadal natural variability of the AMOC is still high. As we do not have additional ensemble members, we do not make strong statements about the impact of historic forcing on the AMOC. However, we can conclude that our simulated AMOC response to historical forcing is consistent with recent synthesis based on observations (Caesar et al., 2022), and the ability for rapid AMOC state changes, as described in Ollinaho et al. (2017), does seem to exist in AWI-CM3, since our low resolution is indeed higher than their high resolution case. While the upper cell diminishes considerably under historic forcing, the lower one is mostly unaffected.

AMOC variability across all latitudes is well correlated, pointing to the same (northern) origin of the signal. In the HIST run the decrease of AMOC is found across all latitudes. The correlation is high but not perfect (see years between 1920 and 1930, for instance). This indicates that the recirculating cell, associated with physical and numerical mixing in the model caused by the advection operator, changes in strength (Sidorenko et al., 2021). Further conclusions would require extra analysis in the density space and the isopycnal framework (as in Sidorenko et al. (2021)) which we didn't activate in this run. However, this extra analysis was done by Sidorenko et al. (2021) and they concluded that the NAO was the main driver of AMOC variability.

Simulated volume transport fluxes were computed across several major ocean straits (Table C2). Historical values averaged over 1985-2014 match well the observation-based estimates for most straits. Two exceptions are the Nares and Davis Straits where the simulated transport is lower than in the observational estimates. Indeed, these very narrow straits are likely not well resolved on the CORE2 mesh. In the Arctic, the poleward inflow through the Barents Sea Opening and Bering Strait is

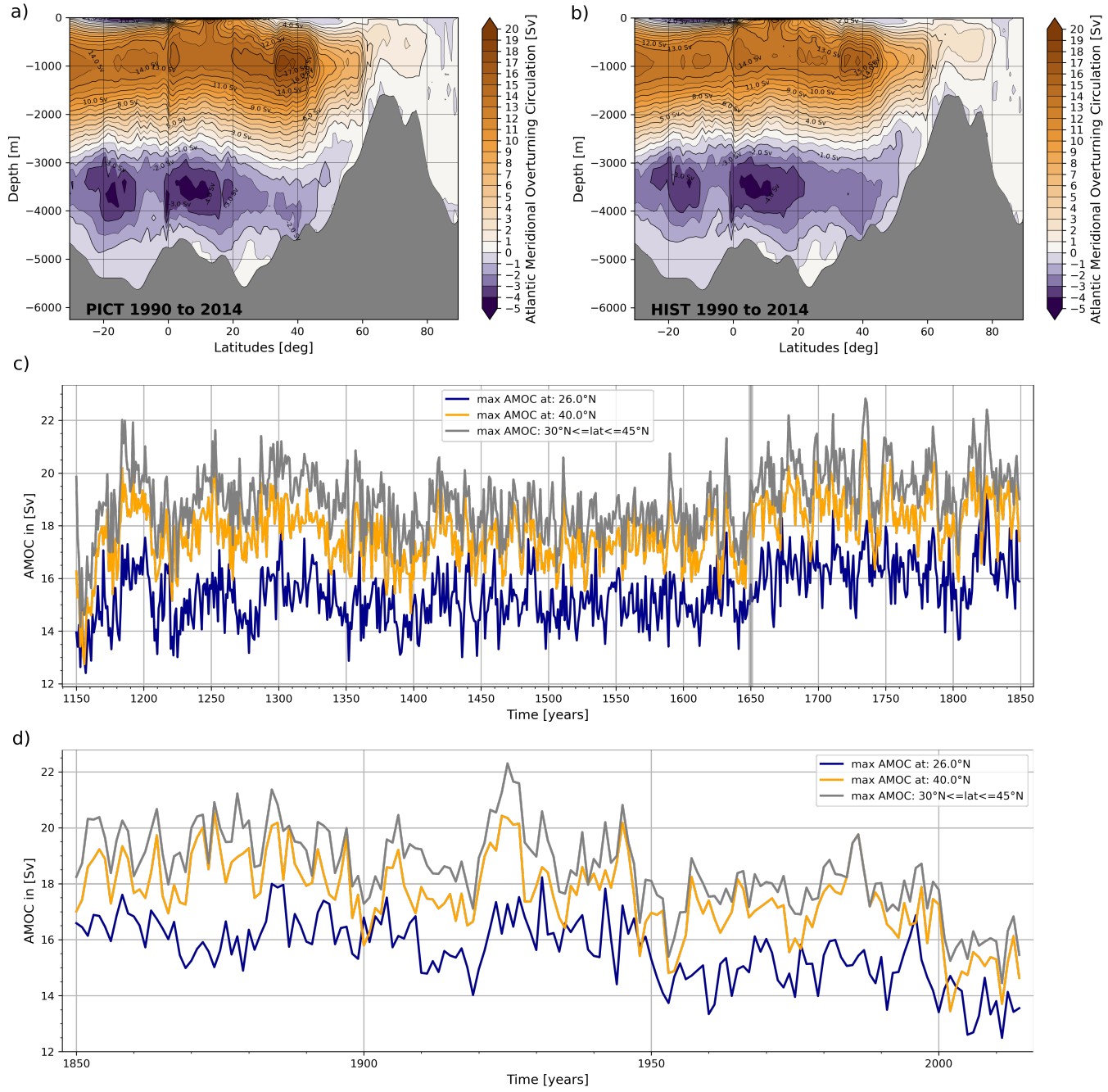

**Figure 13. a)** Atlantic Meridional Overtuning Circulation (AMOC) streamfunction averaged over the last 25 years in the pre-industrial control (PICT) simulation. **b)** as a) but for the historic (HIST) simulation. **c)** Evolution of the AMOC maximum at different latitudes throughout the spinup simulation (SPIN) that precedes both PICT and HIST. **d)** Evolution of the AMOC during HIST.

somewhat overestimated, and counterbalanced by a southward transport at Fram Strait larger than in the observational estimates (yet all remain within the uncertainty range). The simulated Antarctic Circumpolar Current (ACC) transport through the Drake Passage is within the range of observations. While being lower than the most recent measurements by Donohue et al. (2016), our AWI-CM3 estimate is comparable to the CMIP5 multi-model mean (MMM) and larger than the CMIP6 MMM (Beadling et al., 2020).

Historical transports through major straits are thus satisfactorily reproduced in the present AWI-CM3 configuration. Future configurations with an increased resolution, eddy-resolving ocean are expected to provide even more accurate results within narrow straits and in eddy-rich areas such as the Southern Ocean, where mesoscale activity is key in accurately depicting the ACC behaviour (e.g., Rackow et al., 2022).

## 4.5    Climate sensitivity experiments

### 490   4.5.1    Equilibrium Climate Sensitivity (ECS)

To investigate the ability of AWI-CM3 to simulate warmer climate states, we conducted the 4xCO$_2$ experiment which prescribes a sudden permanent increase of the CO2 concentrations to 4 times (1137.27 ppm) the base value (284.32 ppm) from 1850. It can be used to analyse the model's inherent climate sensitivity. As shown by Gregory et al. (2004) the relationship between the change in net downward radiative flux and the change in near surface (2m) air temperature can be described by

a linear regression model. The Gregory plot in Figure 14 a) was computed from the 4xCO$_2$ experiment in comparison to the pre-industrial control simulation. The first axis intersection point determines the instantaneous radiative forcing $F = 7.06 \frac{W}{m^2}$ resulting from the quadrupling of CO2. The linear regression line intersects the second axis at an Equilibrium Temperature difference $\Delta T = 6.49\,^{\circ}C$, resulting in a climate response parameter of AWI-CM3 of $\alpha = \frac{\Delta T}{F} = 0.92 \frac{K}{Wm^2}$. Equilibrium Climate Sensitivity (ECS) is given by doubling rather than quadrupling CO2 concentrations, and is thus ECS$= \frac{\Delta T}{2} = 3.2\,^{\circ}C$. With this

ECS value the AWI-CM3 prototype finds itself near the center of the range predicted by CMIP6 models ($1.8 - 5.6\,^{\circ}C$) (Meehl et al., 2020; Scafetta, 2021), and the same as AWI-CM1, which had a value of $3.2^{\circ}C$ as well.

### 4.5.2    Transient Climate Response (TCR)

We obtain the transient climate response of AWI-CM3 TCo159L91-CORE2 from an experiment that features increased CO$_2$ forcing by 1% per year, starting from the 1850 value of 285 ppm. Radiative forcing thus applied results in a near surface (2m)

air temperature increase of 2.1°C after 70 years, when CO$_2$ concentrations have doubled, as shown in Figure 14b. As with the ECS, the TCR of AWI-CM3 is also near the center of the CMIP6 model distribution of $1.8 - 5.6\,^{\circ}C$ (Scafetta, 2021). Compared to the predecessor model AWI-CM1, the TCR did not change. Interestingly, a further doubling to a total of four times the 1850 CO$_2$ concentrations until the year 1990 results in another 2.6°C increase, and thus a larger global mean near surface (2m) air temperature rise than during the first period.

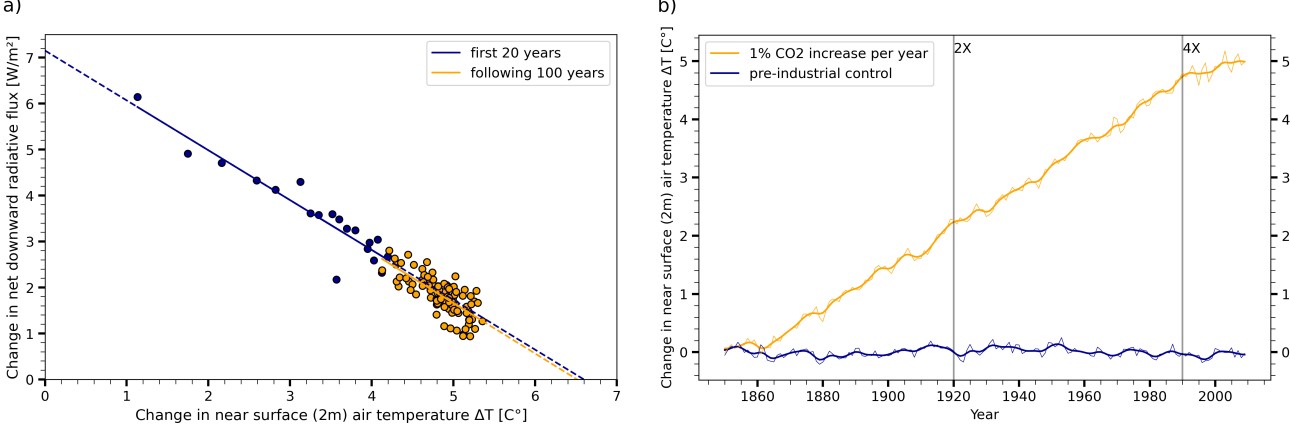

**Figure 14. a)** Gregory regression plot from the abrupt 4xCO$_2$ experiment in comparison to the pre-industrial control. We construct the linear regression of near surface (2m) air temperature ($\Delta T$) change against the downward net radiative flux change ($\Delta F$). From its axis intersection points we obtain the radiative forcing $\Delta F = 7.06 \frac{W}{m^2}$, and the Equilibrium temperature difference $\Delta T = 6.49\,°C$. The climate response parameter of AWI-CM3 is then $\alpha = \frac{\Delta T}{\Delta F} = 0.92 \frac{K}{W}$ and the Equilibrium Climate Sensitivity ECS$= \frac{\Delta T}{2} = 3.24\,°C$. The first 20 years already result in a linear regression almost identical to the following 100 years. **b)** As Figure 11 b), but for the experiment with 1% increase of CO$_2$ per year. Vertical lines indicate doubling and quadrupling of CO$_2$ concentrations. The estimated Transient Climate Response is 2.1°C.

## 5 Computational Performance

The computational performance of a climate model can be measured according to a variety of criteria. Balaji et al. (2017) provide a good overview of what can be considered the computational performance, but in our analysis we will focus on just two aspects, the Simulated Years Per Day (SYPD) and the computational cost measured in Core Hours per Simulated Year (CHSY). Systematic and rigorous experiment design would require that we identify all the degrees of freedom and vary them in all combinations. Unfortunately the number of degrees of freedom is large, including atmosphere and ocean vertical and horizontal resolution, atmospheric spectral and grid point resolution, number of cores allocated for MPI and/or OpenMP parallelization for FESOM2, OpenIFS43 and XIOS, as well as the amount of model output for each of the main components etc. Testing all combinations is impractical. Instead we present results for setups that have been optimized empirically, involving not only the use of analytical tools such as Dr.Hook (Saarinen et al., 2005), LUCIA (Maisonnave et al., 2020) and the XIOS internal statistics, but also educated guesswork. It may well be that better configurations exist, but AWI-CM3 can at a minimum perform to the level presented here.

Tables 1 and 2 list the SYPD and CHSY values we achieved when optimization is done for speed and cost, respectively. Note, that the scaling limit of TCo319L137-DART has not been reached and simulations with upwards of 10 SYPD are likely feasible.

| Atmosphere grid | Ocean mesh | IO scheme | Cores | SYPD | CHSY |
|---|---|---|---|---|---|
| TCo95L91 | CORE2L47 | Sequential | 2593 | 124.05 | 501 |
| TCo95L91 | CORE2L47 | XIOS parallel | 2689 | 134.24 | 480 |
| TCo159L91 | CORE2L47 | Sequential | 2593 | 60.74 | 1024 |
| TCo159L91 | CORE2L47 | XIOS parallel | 2833 | 68.7 | 988 |
| TCo319L137 | DARTL80 | XIOS parallel | 15680 | 7.90 | 47528 |

**Table 1.** Computational performance optimized for integration speed. Simulated Years Per Day (SYPD), and core hours per simulated year (CHSY) of AWI-CM3 for three atmospheric grids, two ocean meshes and two IO schemes are shown. Values were measured on HPC system juwels@fz-juelich.de with Intel Xeon Platinum 8168 CPU, 2× 24 cores, 2.7 GHz processors, accepting sub-linear strong scaling. The atmospheric grids TCo95L91, TCo159L91 and TCo319L137 have a grid point resolution of 100, 61 and 31 km and 91, 91 and 137 vertical layers, respectively. The CORE2 mesh (Figure 2a) has 47 vertical layers, ~127.000 surface nodes, a peak resolution of 20km in Northern mid to high latitudes and up to 125km in Subtropical Gyres. The horizontal resolution of the DART mesh follows the local Rossby radius, peaks at a highest resolution of 5km and has a maximum spacing of 27km, with ∼3.1 million surface nodes and 80 vertical layers.

| Atmosphere grid | Ocean mesh | IO scheme | Cores | SYPD | CHSY |
|---|---|---|---|---|---|
| TCo95L91 | CORE2L47 | XIOS parallel | 721 | 52.34 | 330 |
| TCo159L91 | CORE2L47 | XIOS parallel | 1297 | 42.71 | 736 |
| TCo319L137 | DARTL80 | XIOS parallel | 6769 | 4.61 | 35220 |

**Table 2.** As table 1 but optimized for computational cost instead of integration speed by limiting total core numbers within the limit of linear strong scaling.

## 5.1 High resolution outlook

While we document mainly the first CMIP-prototype simulations of AWI-CM3 and its strengths and weaknesses at low resolution, we also tested higher resolution configurations. In Figure 15 we show the performance of a 31km atmosphere with 137 vertical layers, coupled to a high-resolution ocean with a mesh that features 3.1 million surface nodes and 80 layers. This configuration with the name TCo319L137-DART has been run for 50 years under constant 1990 forcing, with some of the insights gained while performing the set of experiments at low resolution already taken into account.

Key improvements of the TCo319L137-DART setup compared to TCo159L91-CORE2 are the eddy resolving ocean in the western boundary currents, with eddy permittance in the ACC region, as well as better representation of orography and related effects in the atmospheric model. In lower latitude regions the atmosphere has sufficient resolution to resolve and react to ocean eddies. Both atmosphere and ocean feature more vertical layers, allowing for better representation of vertical processes.

The ability of the TCo319L137-DART simulation to reproduce a climate as observed during the years 1990 to 2014 is better than that of the TCo159-CORE2 runs we analysed so far. The improvement is almost universal with only the sea ice

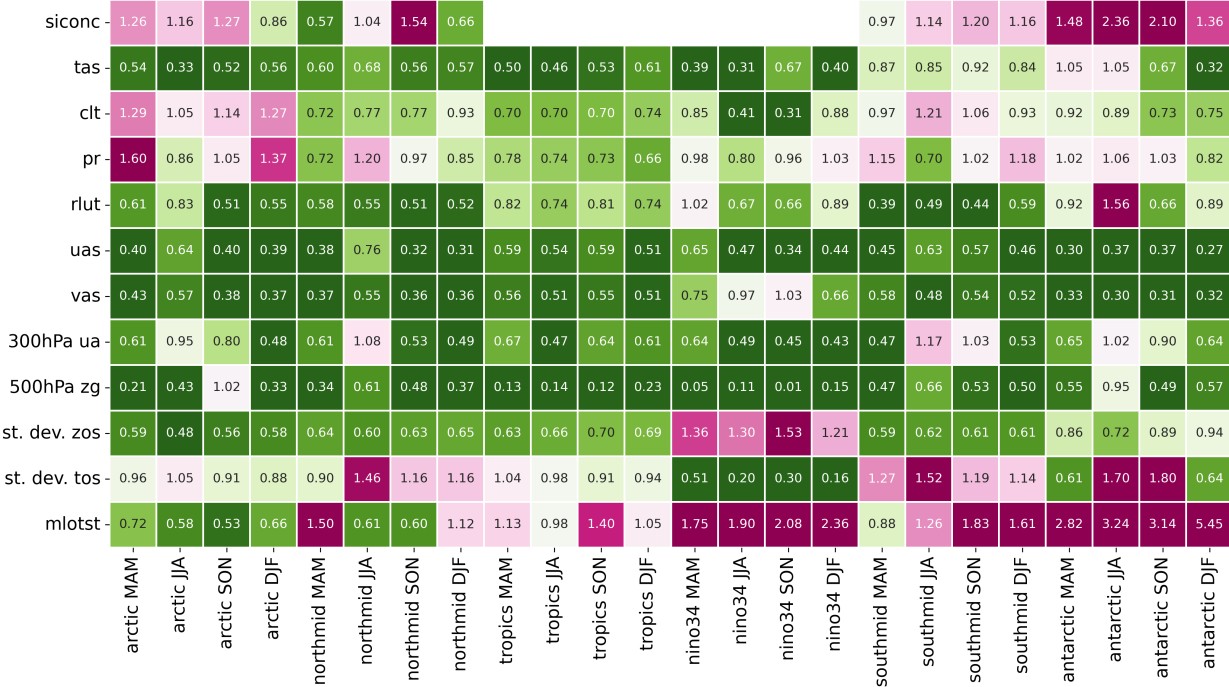

**Figure 15.** Same as figure 4 but for the last 10 years of a 50-year-long TCo319L137-DART simulation under constant 1990 forcing. Biases to observations are not only smaller than in our TCo159L91-CORE2 simulations, but also smaller than those of all but two (HadGEM3MM & NOAA-GFDL) of the CMIP6 models listed in A (Performance indices for CMIP6 models not shown).

concentrations and mixed layer depths still showing below-average performance compared to CMIP6 models. A TCo159-CORE2 simulation of the same length and with the same forcing (not shown) showed performance almost identical to the last 25 years of the HIST experiment discussed above. We conclude that the improvement is related to the model resolution and

continued model development, rather than the shorter run length or different forcing.

Obviously the improved climatological performance comes at a cost, as detailed in Section 5. Every simulated year with TCo319L137-DART costs 35 times the CHSY and is performed at 15 times lower SYPD. More detailed exploration of the higher resolution capabilities of AWI-CM3 will be subject of future work.

## 6  Conclusions

We developed a new coupled climate model AWI-CM3, by coupling the AWI ocean model FESOM2, the ECMWF NWP atmosphere model OpenIFS CY43R3, a small runoff-mapper model, and the XIOS parallel IO library. The coupling exchange is achieved via the OASIS3-MCT 4.0 library.

We ran a set of experiments closely resembling the Coupled Model Intercomparison Project phase 6 (CMIP6) Diagnostic, Evaluation and Characterization of Klima (DECK) simulations to evaluate the representation of the climatological state and the computational performance of the new model. From the experiments we found that, when activating the humidity mass fixer for OpenIFS, the model was able to reach a near equilibrium under constant 1850 well-mixed greenhouse gas and solar forcing. After 700 years of spinup we branched off four experiments, a historic simulation (165y), a pre-industrial control run (165y), and two idealized experiments, one with a sudden increase to $4xCO_2$ (120y) and the other with $1\%$ $CO_2$ (150y) increase per year.

Climate sensitivity experiments with AWI-CM3 obtained an Equilibrium Climate Sensitivity and Transient Climate Response of 3.2°C and 2.1°C, respectively, both of which are near the center of the CMIP6 spreads.

Using the last 25 years of the historical simulation we established that a low resolution version of AWI-CM3 provides above CMIP6-average performance for representing the climatological state of precipitation, wind speeds, cloud fraction, 500 hPa geopotential height and air temperature north of 30°S. We found that the Southern Ocean sea ice concentration and thickness were severely underestimated, leading to large positive near-surface air temperature biases in this region, and traced the sea ice biases to spuriously large mixed layer depth, a positive ice albedo feedback and biases in shortwave downward radiation associated with cloud fraction between 45-60°S.

AWI-CM3 is capable of realistically simulating the Atlantic meridional overturning circulation (AMOC) in terms of both the shape and strength of the streamfunction, as well as reproducing a decreasing trend in the historical period consistent with observations. While the model produces an El Niño-Southern Oscillation (ENSO) at realistic frequencies, the amplitude of ENSO is currently underestimated.

Our recommendations for resolving the Southern Ocean sea ice concentration and thickness biases in future versions of AWI-CM3 include re-tuning of the vertical mixing scheme, and the inclusion of coupling minor fluxes that are at least one magnitude smaller than the ones exchanged so far. We have identified the coupling of ocean surface currents, rain temperature, enthalpy of snow falling into the ocean, enthalpy of melting icebergs and basal melt flux as promising candidates to reduce model biases. Based on literature we suggest porting and activating the Stochastically Perturbed Parametrisations scheme as a potential way to improve ENSO amplitude.

The advanced computational efficiency and scalability of AWI-CM3, combined with a very solid model and coupling physics implementation, will eventually enable us to perform full DECK and scenario simulations at resolutions of 5-25km, that were previously reserved for the high end of the HighResMIP protocol. We provide a preview with a shorter high-resolution simulation, indicating that most AWI-CM3 climatological biases at future operational resolution will be about half those of the average CMIP6 model.

*Code and data availability.* The ocean model FESOM2 source code is available on Zenodo at $10.5281/zenodo.6335383$ and at $https://github.com/FESOM/fesom2/releases/tag/AWI-CM3\_v3.0$. OpenIFS is not publicly available but rather subject to licecing by ECMWF. However licences are readily given free of charge to any academic or research institute. All modifications required to en-

able AWI-CM3 simulations with OpenIFS CY43R3V1 as provided by ECMWF can be obtained on Zenodo at: $10.5281/zenodo.6335498$. The OASIS coupler is available upon registration at: https://oasis.cerfacs.fr/en/downloads/. The XIOS source code is available on Zenodo ($10.5281/zenodo.4905653$ Meurdesoif, 2017) and on the official repository ($http://forge.ipsl.jussieu.fr/ioserver$, last access: 4 March 2022). The runoff mapper scheme is available on Zenodo at $10.5281/zenodo.6335474$. The compile and runtime engine esm-tools is available on Zenodo at: $10.5281/zenodo.6335309$. All data required to reproduce the plots shown here can be found at: $10.5281/zenodo.6337593$, $10.5281/zenodo.6337571$, and $10.5281/zenodo.6337627$. The processing and plotting scripts for the reproduction of the shown analysis can be found at: $10.5281/zenodo.6335530$. Documentation of AWI-CM3 and a user guide can be found at: $awi - cm3 - documentation.readthedocs.io$

**Appendix A:  List of CMIP6 models for climate model performance index calculation**

ACCESS-CM2, AWI-CM-1-1-MR, BCC-CSM2-MR, CAMS-CSM1-0, CAS-ESM2-0, CanESM5, CIESM, CESM2, CMCC-CM2-SR5, CNRM-CM6-1-HR, FGOALS-f3-L, FIO-ESM-2-0, E3SM-1-1, EC-Earth3, GFDL-CM4, GISS-E2-1-G, HadGEM3-GC31-MM, ICON-ESM-LR, IITM-ESM, INM-CM5-0, IPSL-CM6A-LR, KIOST-ESM, NESM3, NorESM2-MM, MCM-UA-1-0, MIROC6, MPI-ESM1-2-LR, MRI-ESM2-0, SAM0-UNICON, TaiESM1

## Appendix B: Additional figures

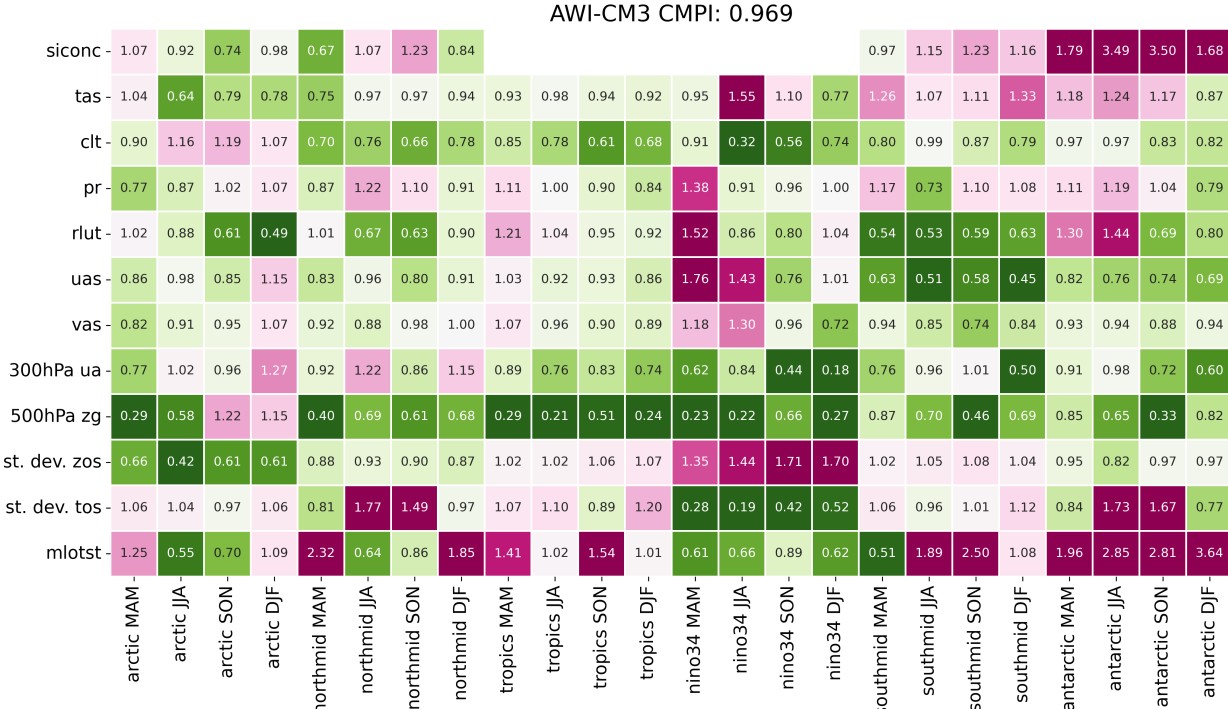

**Figure B1.** HIST biases as figure 4 , but using NCEP-DOE2 re-analysis by Kistler et al. (2001) instead of ERA5 for near-surface air temperature (tas), eastward near-surface wind (uas), northward near-surface wind (vas), 300 hPa eastward wind (ua), 500 hPa geopotential height. Note that ERA5 was created using IFS CY41R2, a model closely related to the OpenIFS CY43R3 atmosphere employed in AWI-CM3.

# AWI-CM3-DART CMPI: 0.893

| | arctic MAM | arctic JJA | arctic SON | arctic DJF | northmid MAM | northmid JJA | northmid SON | northmid DJF | tropics MAM | tropics JJA | tropics SON | tropics DJF | nino34 MAM | nino34 JJA | nino34 SON | nino34 DJF | southmid MAM | southmid JJA | southmid SON | southmid DJF | antarctic MAM | antarctic JJA | antarctic SON | antarctic DJF |
|---|---|---|---|---|---|---|---|---|---|---|---|---|---|---|---|---|---|---|---|---|---|---|---|---|
| siconc | 1.26 | 1.16 | 1.27 | 0.86 | 0.57 | 1.04 | 1.54 | 0.66 | | | | | | | | | 0.97 | 1.14 | 1.20 | 1.16 | 1.48 | 2.36 | 2.10 | 1.36 |
| tas | 0.63 | 0.83 | 0.95 | 0.57 | 0.83 | 0.82 | 0.83 | 0.84 | 0.93 | 0.94 | 0.91 | 0.94 | 1.10 | 0.89 | 1.09 | 1.30 | 0.84 | 0.95 | 0.98 | 0.83 | 0.91 | 0.97 | 0.92 | 0.73 |
| clt | 1.29 | 1.05 | 1.14 | 1.27 | 0.72 | 0.77 | 0.77 | 0.93 | 0.70 | 0.70 | 0.70 | 0.74 | 0.85 | 0.41 | 0.31 | 0.88 | 0.97 | 1.21 | 1.06 | 0.93 | 0.92 | 0.89 | 0.73 | 0.75 |
| pr | 1.60 | 0.86 | 1.05 | 1.37 | 0.72 | 1.20 | 0.97 | 0.85 | 0.78 | 0.74 | 0.73 | 0.66 | 0.98 | 0.80 | 0.96 | 1.03 | 1.15 | 0.70 | 1.02 | 1.18 | 1.02 | 1.06 | 1.03 | 0.82 |
| rlut | 0.61 | 0.83 | 0.51 | 0.55 | 0.58 | 0.55 | 0.51 | 0.52 | 0.82 | 0.74 | 0.81 | 0.74 | 1.02 | 0.67 | 0.66 | 0.89 | 0.39 | 0.49 | 0.44 | 0.59 | 0.92 | 1.56 | 0.66 | 0.89 |
| uas | 0.77 | 0.90 | 0.82 | 0.82 | 0.67 | 0.97 | 0.71 | 0.69 | 0.73 | 0.79 | 0.77 | 0.75 | 0.96 | 0.81 | 0.29 | 0.71 | 0.65 | 0.82 | 0.76 | 0.67 | 0.82 | 0.79 | 0.78 | 0.75 |
| vas | 0.84 | 0.91 | 0.83 | 0.85 | 0.75 | 0.83 | 0.84 | 0.89 | 0.81 | 0.88 | 0.86 | 0.80 | 0.75 | 0.92 | 1.15 | 0.84 | 0.87 | 0.79 | 0.81 | 0.87 | 0.94 | 0.92 | 0.96 | 0.99 |
| 300hPa ua | 0.58 | 0.94 | 0.76 | 0.50 | 0.59 | 1.08 | 0.50 | 0.51 | 0.61 | 0.51 | 0.64 | 0.63 | 0.52 | 0.46 | 0.51 | 0.40 | 0.44 | 1.21 | 1.04 | 0.55 | 0.65 | 1.21 | 0.97 | 0.50 |
| 500hPa zg | 0.25 | 0.43 | 0.93 | 0.44 | 0.43 | 0.47 | 0.30 | 0.46 | 0.23 | 0.26 | 0.17 | 0.37 | 0.08 | 0.11 | 0.03 | 0.19 | 0.40 | 0.72 | 0.56 | 0.44 | 0.77 | 1.25 | 0.78 | 0.62 |
| st. dev. zos | 0.59 | 0.48 | 0.56 | 0.58 | 0.64 | 0.60 | 0.63 | 0.65 | 0.63 | 0.66 | 0.70 | 0.69 | 1.36 | 1.30 | 1.53 | 1.21 | 0.59 | 0.62 | 0.61 | 0.61 | 0.86 | 0.72 | 0.89 | 0.94 |
| st. dev. tos | 0.96 | 1.05 | 0.91 | 0.88 | 0.90 | 1.46 | 1.16 | 1.16 | 1.04 | 0.98 | 0.91 | 0.94 | 0.51 | 0.20 | 0.30 | 0.16 | 1.27 | 1.52 | 1.19 | 1.14 | 0.61 | 1.70 | 1.80 | 0.64 |
| mlotst | 0.72 | 0.58 | 0.53 | 0.66 | 1.50 | 0.61 | 0.60 | 1.12 | 1.13 | 0.98 | 1.40 | 1.05 | 1.75 | 1.90 | 2.08 | 2.36 | 0.88 | 1.26 | 1.83 | 1.61 | 2.82 | 3.24 | 3.14 | 5.45 |

**Figure B2.** TCo319L137-DART biases as figure 15, but using NCEP-DOE2 re-analysis by Kistler et al. (2001) instead of ERA5 for near-surface air temperature (tas), eastward near-surface wind (uas), northward near-surface wind (vas), 300 hPa eastward wind (ua), 500 hPa geopotential height. Note that ERA5 was created using IFS CY41R2, a model closely related to the OpenIFS CY43R3 atmosphere employed in AWI-CM3.

 **Appendix C: Table of observational datasets for climate model performance index calculation**

| Variable | Longname | Dataset | Time range |
|---|---|---|---|
| tas | Near surface (2m) air temperature | ERA5 Reanalysis | 1989/11/01 to 2014/11/30 |
| uas | Near surface (10m) zonal wind speed | ERA5 Reanalysis | 1989/11/01 to 2014/11/30 |
| vas | Near surface (10m) meridional wind speed | ERA5 Reanalysis | 1989/11/01 to 2014/11/30 |
| 300 hPa ua | 300 hPa zonal wind speed | ERA5 Reanalysis | 1989/11/01 to 2014/11/30 |
| 300 hPa zg | 500 hPa geopotential height | ERA5 Reanalysis | 1989/11/01 to 2014/11/30 |
| pr | Precipitation flux | GPCP v2.3 | 1989/11/01 to 2014/11/30 |
| siconc | Sea ice area fraction | OSISAF OSI-450 | 1989/11/01 to 2014/11/30 |
| rlut | TOA outgoing longwave flux | CERES-EBAF | 2000/03/15 to 2014/06/30 |
| clt | Cloud area fraction | MODIS Atmosphere L2 | 2000/03/15 to 2014/11/30 |
| Std Dev tos | Std Dev of ocean surface temperature | HadISST | 1989/11/01 to 2014/11/30 |
| Std Dev zos | Std Dev of sea surface height | JASON-1, JASON-2, CryoSat | 2002/01/01 to 2014/11/30 |
| mlotst | Mixed layer depth | C-GLORSv7 Reanalysis | 1989/11/01 to 2014/11/30 |

**Table C1.** Table of observational datasets for climate model performance index calculation

| Transport (Sv) | AWI-CM3 HIST | Observations | References of observations |
|---|---|---|---|
| Fram Strait | -2.96 | $-2.0 \pm 2.7$ | Schauer et al. (2008) |
| Davis Strait | -0.42 | $-1.6 \pm 0.5$ | Curry et al. (2014) |
| Bering Strait | 1.19 | $0.83 \pm 0.66, 1.0 \pm 0.05$ | Roach et al. (1995), Woodgate (2018) |
| Nares Strait | -0.31 | $-0.57 \pm 0.09, -0.8 \pm 0.3$ | Münchow and Melling (2008), Münchow et al. (2006) |
| Barents Sea Opening | 2.46 | 2.0 | Smedsrud et al. (2010) |
| Drake Passage | 148.63 | $136.7 \pm 6.9, 173.3 \pm 10.7$ | Meredith et al. (2011), Donohue et al. (2016) |
| Mozambique Channel | -19.59 | $-16 \pm 8.9$ | Ridderinkhof et al. (2010) |

**Table C2.** Transport fluxes [Sv] through a number straits and channels as simulated by AWI-CM3 in comparison to observational estimates. Analysis period for was 1990-2014 of the historic simulation.

*Author contributions.* J. Streffing: Conceptualization, coupling methodology, coding, low resolution numerical experiments, analysis, and writing original draft preparation; D. Sidorenko, J. Kjellsson, U. Fladrich and L. Zampieri: Coupling methodology, coding; N. Koldunov, D. Sidorenko and T. Rackow: development of analysis methods for unstructured grids; H. Goessling, Q. Wang, L. Mu and S. Danilov, T. Jung: Conceptualization; T.Semmler and D. Sein: high resolution numerical experiments; J. Hegewald: High resolution computational and IO optimization; M. Athanase: Analysis and writing original draft of transports Section; M. Andrés-Martínez, D. Barbi and P. Gierz: compile, runtime optimization; S. Danilov, S. Juricke, G. Lohmann, and T. Jung: supervision, funding acquisition. All authors discussed, read, edited, and approved the article. All authors have read and agreed to the published version of the manuscript.

*Competing interests.* At least one of the (co-)authors is a member of the editorial board of Geoscientific Model Development.

*Acknowledgements.* This paper is a contribution to the projects L4, S1 and S2 of the Collaborative Research Centre TRR 181 "Energy Transfers in Atmosphere and Ocean" funded by the Deutsche Forschungsgemeinschaft (DFG, German Research Foundation) under Project 274762653. We thank the Jülich Supercomputing Centre for providing a share of the JUWELS ESM Partition under the compute projects chhb19, chhb20 and cesmtst as well as the data storage project hhb19. The authors gratefully acknowledge the Gauss Centre for Supercomputing e.V. (www.gauss-centre.eu) for funding this project by providing computing time through the John von Neumann Institute for Computing (NIC) on the GCS Supercomputer JUWELS at Jülich Supercomputing Centre (JSC). We also thank the Helmholtz ESM-Project for technical support. H. Goessling and M. Athanase acknowledge funding by the Federal Ministry of Education and Research of Germany in the framework of SSIP (grant01LN1701A). D. Sein acknowledge funding by the Federal Ministry of Education and Research of Germany (BMBF) in the framework of ACE (grant 01LP2004A) and by the Ministry of Science and Higher Education of Russia (theme No. FMWE-2021-0014). J. Kjellsson acknowledges funding by the ROADMAP project from JPI Oceans & JOI Climate (grant 01LP2002C).

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
