# Peer review of "AWI-CM3 coupled climate model: Description and evaluation experiments for a prototype post-CMIP6 model"

_EGUsphere, 2022_

## Referee Comment (RC1)

This study describes a new version of the AWI-CM3 and shows that it has good skills in representing the observed climatology and better computational efficiency compared to previous versions. Also, authors discuss model biases and give some suggestions to make further improvement. The development of new version is very impressive and represents the cutting edge of Earth system modeling. The results are noteworthy and can potentially help understand model biases. I believe this manuscript provides a useful roadmap for ongoing challenge to develop next generation of Earth system model including development of high-resolution climate model. I recommend publishing this paper after some minor revisions.

Suggested improvements are as follows:

1. Evaluation of model performance.
Authors calculated climate model performance indices based on the ideas of Reicher and Kim (2008). What about temporal variability? Does this method consider the variability? If not, I would suggest adding more analysis on climate variability. Regarding the interannual variability authors show ENSO timeseries only. Spatial patten and amplitude of interannual variability as well as seasonal cycle are also important to evaluate model's performance.

2. Figure 1 and Table 1: It would be great if authors can combine the information in Fig.1 and Table 1. But, if it looks too busy, ignore my suggestion.

3. Figure 5 and its description: Please add plots for observations to compare spatial distribution and temporal evolution of sea ice. And, what is the dotted lines in Fig. 5i?

4. I don't think Fig. 9a is necessary. Instead, it would be great if authors show spatial patten of AWI-CM3 SST variability. Does this model show seasonal locking of ENSO?

5. Figure 6: I don't see any description for Fig. 6c.

6. Figure 7: I guess this is annual mean climatology. Please specify clearly.

7. Figure 7 & 11 label bar: To describe the results precisely, authors should indicate each values of color bar since the scale does not increase/decrease uniformly.

8. Section 2.4: I think XIOS works for OpenIFS only. Then, is there any IO scheme for FESOM?

9. Line 104: I'm wondering if the ocean basins with narrow outflow can be included in the DART resolution.

10. Line 225: The list of observational datasets used to calculate all mean absolute errors is also given in Appendix B -> Appendix C?

11. Line 320: If this cold bias in the IO is likely due to inability to resolve narrow strait on the

FESOM2 CORE2 mesh, then can we see some improvements of the performance in the high-resolution DART simulation?

12. Line 325: Biases presented in 8 explain -> Biases presented in Figure 8 explain

13. Line 388: 20 Sv along with the cooling of the upper ocean (Figure 3c) -> Figure 3b

14. Line 424: "m$^2$" should be included in the unit of $\alpha$.

15. Line 433: What is the TCR of AWI-CM1?

16. Line 461: What is main model improvement TCo319L137-DART compared to TCo159-CORE2.

17. Figure B1 should be Appendix D.

18. Please check use of abbreviation in the text.
E.g. Line 14: The evolution of coupled climate models between phases of the Coupled Model Intercomparison Project (CMIP) is advancing
Line 470: We ran a set of experiments closely resembling the Coupled Model Intercomparison Project phase 6 (CMIP6) DECK -> We ran a set of experiments closely resembling the CMIP6 DECK
Line 439-440: the Simulated Years Per Day (SYPD) and the computational cost measured in Core Hours per Simulated Year (CHSY)
Line 448: Tables 2 and 3 list the Simulated Years Per Day (SYPD) and core hours per simulated year (CHSY) values -> Tables 2 and 3 list the SYPD and CHSY values

19. Please check reference list.
E.g. Line 580 and 588.

---

## Author Comment (AC1)

**Response to reviewer #1**

This study describes a new version of the AWI-CM3 and shows that it has good skills in representing the observed climatology and better computational efficiency compared to previous versions. Also, authors discuss model biases and give some suggestions to make further improvement. The development of new version is very impressive and represents the cutting edge of Earth system modeling. The results are noteworthy and can potentially help understand model biases. I believe this manuscript provides a useful roadmap for ongoing challenge to develop next generation of Earth system model including development of high resolution climate model. I recommend publishing this paper after some minor revisions.

> *We would like to thank the reviewer for the encouraging and very positive feedback.*

Suggested improvements are as follows:

1. Evaluation of model performance.

Authors calculated climate model performance indices based on the ideas of Reicher and Kim (2008). What about temporal variability? Does this method consider the variability? If not, I would suggest adding more analysis on climate variability. Regarding the interannual variability authors show ENSO timeseries only. Spatial patten and amplitude of interannual variability as well as seasonal cycle are also important to evaluate model's performance.

> *The method of Reichler and Kim (2008) can be used to consider basic climate variability. To do so we can infer the climate variability from both the observational data, as well as from CMIP6 and AWI-CM3. We have added the standard deviation of sea surface temperature and sea surface height as general measures of model variability. For both the existing and newly added variables in the performance index we show biases with seasonal time resolution.*
>
> *Furthermore we added mixed layer depth as one of the key areas of future work on AWI-CM3.*
>
> *We also updated the list of CMIP6 model simulations against which we normalize our model performance (see: https://github.com/JanStreffing/cmpi-tool/issues/1). We ensured that for each model family we use only the highest resolution, lowest complexity (in terms of ESM-components) version. In doing so the CMIP6 ensemble considered has become somewhat more skillful, leading to a slight drop in the relative performance of our AWI-CM3 prototype.*

*Reichler, Thomas, and Junsu Kim. "How well do coupled models simulate today's climate?." Bulletin of the American Meteorological Society 89.3 (2008): 303-312.*

2. Figure 1 and Table 1: It would be great if authors can combine the information in Fig.1 and Table 1. But, if it looks too busy, ignore my suggestion.

*We agree with the suggestion and combined Figure 1 and Table 1.*

3. Figure 5 and its description: Please add plots for observations to compare spatial distribution and temporal evolution of sea ice. And, what is the dotted lines in Fig. 5I?

*We added GIOMAS reanalysis for the spatial distribution of sea ice thickness. We believe that the sea ice extent plot is already quite busy, and that the thickness reanalysis gives a good overview of the extent match/mismatch. The dotted lines are merely mean as visual aid for a plot that is rather wide.*

4. I don't think Fig. 9a is necessary. Instead, it would be great if authors show spatial patten of AWI-CM3 SST variability. Does this model show seasonal locking of ENSO?

*We have modified Fig. 9a such that it shows the EOF1 as correlation based on:* https://ajdawson.github.io/eofs/latest/userguide/overview.html#eof-analysis-with-eofs

*We have worked on ENSO phase locking in the 6 months since we ran the simulations upon which this paper is based. Initially phase locking was found to be weak. After improving the near equators SST and Precipitations biases phase locking looks better in newer versions of the model. We added one sentence about the weak phase locking in the model versions we present here.*

*Since this work is outside the scope of the project and partially by non co-authors, we do not add it into the paper. Here is a preliminary plot for the reviewer showing ongoing work on ENSO phase locking:*

[Figure]

Niño3.4 standard deviation (Monthly)

5. Figure 6: I don't see any description for Fig. 6c.

*We have added the missing descriptions for Fig. 6c.*

6. Figure 7: I guess this is annual mean climatology. Please specify clearly.

*We have clarified that the plots indeed show annual mean biases*

7. Figure 7 & 11 label bar: To describe the results precisely, authors should indicate each values

of color bar since the scale does not increase/decrease uniformly.

*The color bar is symmetric around 0 and omits every other value. The values that have no values on the positive side have one on the negative side and vice versa.*

8. Section 2.4: I think XIOS works for OpenIFS only. Then, is there any IO scheme for FESOM?

*We have added a sentence on the bespoke parallel IO scheme included in FESOM2*

9. Line 104: I'm wondering if the ocean basins with narrow outflow can be included in the

DART resolution.

*Indeed the DART mesh with its resolution down to 4km does include many of these narrow outflow channels. We have added a sentence that states this explicitly.*

10. Line 225: The list of observational datasets used to calculate all mean absolute errors is

also given in Appendix B -> Appendix C?

*There seems to be something amiss with the appendix naming. We think this is a problem with the GMD latex template. We swapped the figures and table around until the Apendix letter matched the positions. Hopefully this will be double checked during typesetting.*

11. Line 320: If this cold bias in the IO is likely due to inability to resolve narrow strait on the

FESOM2 CORE2 mesh, then can we see some improvements of the performance in the highresolution DART simulation?

*First results from shorter (100y long) simulations do indicate this. We consider this a bit outside the scope of this paper, as the high resolution model is still being tuned.*

12. Line 325: Biases presented in 8 explain -> Biases presented in Figure 8 explain

*Fixed.*

13. Line 388: 20 Sv along with the cooling of the upper ocean (Figure 3c) -> Figure 3b

*From the context of the sentence, we were talking about the change in the time series once the total mass fixer was turned on. This timeseries is Figure 3c*

14. Line 424: "m2" should be included in the unit of $\alpha$.

*Fixed.*

15. Line 433: What is the TCR of AWI-CM1?

*The TCR of AWI-CM1 is in fact the same as that of AWI-CM3, barring rounding errors (2.1°C). We rewrote this sentence to express this.*

16. Line 461: What is main model improvement TCo319L137-DART compared to TCo159-CORE2.

*We have added a few sentences outlining the major improvements stemming from increased model resolution.*

17. Figure B1 should be Appendix D.

*See #10*

18. Please check use of abbreviation in the text.

E.g. Line 14: The evolution of coupled climate models between phases of the Coupled Model Intercomparison Project (CMIP) is advancing

Line 470: We ran a set of experiments closely resembling the Coupled Model Intercomparison Project phase 6 (CMIP6) DECK -> We ran a set of experiments closely resembling the CMIP6 DECK

*We double checked our abbreviations. We decided to introduce abbreviations in the conclusion independently from the main text, as readers will frequently skip the text on the first skim through.*

Line 439-440: the Simulated Years Per Day (SYPD) and the computational cost measured in Core Hours per Simulated Year (CHSY)

Line 448: Tables 2 and 3 list the Simulated Years Per Day (SYPD) and core hours per simulated year (CHSY) values -> Tables 2 and 3 list the SYPD and CHSY values

*Fixed*

---

## Author Comment (AC2)

Referee comment on "AWI-CM3 coupled climate model: Description and evaluation experiments for a prototype post-CMIP6 model" by Jan Streffing et al., EGUsphere, https://doi.org/10.5194/egusphere-2022-32-RC2, 2022

This manuscript describes the newly developed climate model AWI-CM3. The global coupled climate model consists of the atmosphere model OpenIFS and the ocean model FESOM2 which are coupled using the OASIS MCT4 coupling software. For the AWI-CM family the new development is mainly the replacement of the ECHAM atmosphere, which was used in AWI-CM2, by OpenIFS. The authors describe the component model and assess the performance by evaluating a set of CMIP6 DECK-like experiments in comparison with observational data sets and other CMIP6 models. This is a standard procedure and the evaluation of the model at moderate resolution shows good and above-CMIP6 standard performance. The authors also comment on computational performance and conclude that this model is quite suitable for CMIP6-like experiments. Finally, they give some hints on the performance of a higher-resolution version.

First of all, achieving such good performance in an early stage of development is a great success and I congratulate the authors. As both, the ocean and atmosphere components have potential to work even better at higher resolution and using more of the flexible grid properties of FESOM, I expect to see interesting configurations as coupled climate model and full Earth System Model in the future.

Overall, the paper covers all relevant aspects of a model documentation and the presentation is mostly clear and concise. However, as I outline below, another iteration seems to be necessary. I feel that important information is missing at some places and the evaluation could be more quantitative at other sections. Often the reader would need to consult other publications for very basic information. Also, the text needs another

revision. For example, acronyms and abbreviations used either without spelling out what they stand for or with their definitions given later in the manuscript elsewhere.

I therefore recommend that the paper should be accepted for publication after taking into account the points below and including further discussions on various points. I would rate the revisions needed somewhere between "minor" and "major".

> *We would like to thank the reviewer for the encouraging feedback.*

Specific comments:

Introduction

General: I recommend to discuss where AWI-CM3 stands in comparison to other recent developments for CMIP6, but also beyond (e.g. new grid systems in MPAS, FESOM, ICON, improved dynamical cores, etc.).

> *We added some additional information about MPAS, new grids and new core design.*

Ln 16ff: the reader may not be familiar with the differences between AWI-CM and AWIESM models. This should be briefly explained.

> *We added brief explanations where these models are introduced.*

Ln 22: be more specific of which range of resolution you are talking about.

Section 2

> *We added a clarification stating that we are talking about the range from current CMIP6 to NWP models (~100->~10km)*

Ln 56 ff and figure 1: are WAM and H-TESSEL actually used here? Later you say that another hydrology model (mHM) shall be introduced later. The description of the atmosphere model should also include some information how land processes are treated.

> *We added a description of the two way WAM ↔ OpenIFS coupling. We also clarify that WAM is currently not directly coupled to FESOM2.*

*We added information about the column model nature of H-TESSEL soil moisture updates, and the need for a Runoff-mapper for horizontal water transport on land. It is this water transport that may be replaced with a more sophisticated scheme.*

Ln 76: what is the vertical lay-out in the atmosphere, which coordinate is used, how is the stratosphere resolved?

*We added information about the vertical layout.*

Ln 82ff: give some more details on FESOM2: e.g. physical paramterizations, like vertical mixing or eddy-induced (GM) mixing

*We added information about GM, KPP, the viscosity closure, & MOMIX.*

Ln 87: a few words more about the sea-ice model, what kind of dynamics and thermodynamics are used in FESIM, how is it coupled to FESOM?

*We added a section of FESIM and how it is integrated into FESOM, and the potential upgrade option Icepack.*

Ln 91: what is the vertical distribution of the levels, how is the mixed layer resolved?

*We added a sentence on level distribution in the upper 100 meters.*

Ln 113: see above

*Answered above.*

Section 3

Ln 144ff and Table 1: In the introduction you quote Renault et al (2016) and say that "local energy transfer" is important. Why didn't you include then the coupling of ocean surface currents for the calculation of the wind stresses?

*While we have recently identified this flux as important, this was not the case at the onset of the project. Neither of the predecessor CMIP6 models AWI-CM1 or EC-Earth3 featured*

*ocean surface current coupling. Furthermore the interface to integrate the ocean surface current into OpenIFS was only recently provided by Barcelona Supercomputing Center.*

*The feature will be added in the next minor release of AWI-CM3.*

Ln 157: I don't understand what "converged solution" means here

*Added explanation that this refers to the theoretical option of using an iterative solver for the ocean-atmosphere surface update.*

Section 4

Ln 175: How is the initial state of the ocean defined? Did you run ocean stand-alone simulations before coupling or start with climatology (which)? From Fig3, I assume it is PHC, but you should describe it in the text.

*Added a sentence on the initialization with PHC3.*

Fig 3: Define PHC

*Definition added.*

Line 197: define KPP

*Definition added.*

Ln 214: As your models runs at above 60SYPD, why don't you provide a control run of decent length (I think CMIP6 asks for 500 years). This would allow you to asses aspects of low-frequency variability (e.g. AMOC, AMV, sea ice extent).

*Indeed we did not follow the CMIP6 DECK protocol here. If we were to repeat the work, we likely would. This was mainly motivated by the fact that we consider this a prototype version. This is not a model with e.g. cmorized output and upload to ESGF. It is however a version that some of our colleagues have started to put to limited scientific use, motivating us to document its features and calling it release 3.0.*

Fig. 4: "Mean absolute error…" aren't all these "relative" errors?

*We reformulated this sentence to hopefully better express what the Kim and Reichelt performance indices are:*

*Performance indices after Reichler and Kim (2008) that give the fraction of absolute error of climatology of the last 25 years in AWI-CM3 historic simulation to the absolute error averaged over CMIP6 models.*

*Reichler, Thomas, and Junsu Kim. "How well do coupled models simulate today's climate?." Bulletin of the American Meteorological Society 89.3 (2008): 303-312.*

Ln 221ff: the Reichler indices are fine, but I would like to see at least a few vertical plots

for the atmosphere, e.g. zonal averages of zonal winds and temperature to see how the

models performs in the upper troposphere/lower stratosphere.

*We added a figure showing vertical plots of zonal mean temperature and zonal mean zonal wind. The latter showed obvious issues in the QBO region, encouraging us to add a QBO plot as well. From this QBO plot a clear issue of the model was found, that we recommend shall be tuned in an upcoming release of the model. We found that other of OpenIFS users had found similar issues and already identified some tuning parameters:*
*https://confluence.ecmwf.int/pages/viewpage.action?pageId=222481088*

Ln 248, Figure 5: include an estimate from observations, e.g. sea-ice extent

*We added GIOMAS reanalysis for the spatial distribution of sea ice thickness. We believe that the sea ice extent plot is already quite busy, and that the thickness reanalysis gives a good overview of the extent match/mismatch.*

Ln 275, Figures 7,8: a more quantitative evaluation could be done including RMS and

mean errors.

*We added area weighted rmsd and md for figues 7 & 8.*

Ln 307: what kind of work on the mixing schemes would help?

*We have since found some indication that replacing MOMIX with a salt plume parameterization might work. However, we think there are still too many issues with the approach to put this into writing.*

Ln 315: are there plans to combine the Langmuir-associated mixing with WAM, as in

CESM2 (see Danabasoglu et al., JAMES, 2020).

*We added a note on our plans to include Langmuir circulation parameterization.*

Ln 320: any idea what causes the strong warm bias in the Atlantic Subpolar Gyre?

*We added the following section outlining different attempts to explain this fairly common model bias:*

*In Sidorenko et al. 2014 we noted a similar bias when we analyzed a coupled setup with FESOM1.0 coupled to ECHAM6.3. This bias is shared between many climate models which contributed to CMIP. A similar drift in ocean hydrography is also described in Sterl et al. (2012), Delworth et al. (2006, 2012), and Jungclaus et al. (2013). These authors discuss different factors that may be responsible for the bias. Sterl et al. (2012) show that overestimation of the Mediterranean outflow can significantly increase the deep-ocean salinity bias. Delworth et al. (2012) attribute this anomaly to the insufficient eddy transport required to compensate for the wind-driven subduction in the subtropical gyres. They show that moving towards an eddy-resolving setting or a parameterization of the eddy stirring reduces the temperature biases significantly. Jungclaus et al. (2013) suggest that part of the problem arises from the improper interbasin exchange between the Indian and South Atlantic oceans.*

*Sidorenko, D., Rackow, T., Jung, T. et al. Towards multi-resolution global climate modeling with ECHAM6–FESOM. Part I: model formulation and mean climate. Clim Dyn 44, 757–780 (2015). https://doi.org/10.1007/s00382-014-2290-6*

*Sterl A, Bintanja R, Brodeau L, Gleeson E, Koenigk T, Schmith T, Semmler T, Severijns C, Wyser K, Yang S (2012) A look at the ocean in the EC-Earth climate model. Clim Dyn 39(11):2631–2657*

*Delworth TL et al (2006) GFDL's CM2 Global coupled climate models. Part I: formulation and simulation characteristics. J Clim 19:643–674*

*Delworth TL et al (2012) Simulated climate and climate change in the GFDL CM2.5 high-resolution coupled climate model. J Clim 25:2755–2781*

*Jungclaus JH, Fischer N, Haak H, Lohmann K, Marotzke J, Matei D, Mikolajewicz U, Notz D, von Storch JS (2013) Characteristics of the ocean simulations in the Max Planck Institute Ocean Model (MPIOM) the ocean component of the MPI-Earth system model. J Adv Model Earth Syst 5(2):422–446*

Ln 329 ff: The section on variability could be extended a bit. At least some spatial

regressions of ENSO could be included. ENSO is not the only mode of variability; several

recent papers on CMIP6 models (e.g., Voldoire et al., 2019; Danabasoglu et al., 2020)

mhave included, for example MJO, which is quite revealing for the atmosphere.

*We have added the spatial regression of ENSO and information about QBO.*

*Unfortunately our outgoing thermal radiation, the standard field for plotting MJO was only stored at monthly frequency. Too coarse for MJO detection or a Wheeler Kiladis plot.*

Ln 379: here as well, I would expect a little bit more quantitative evaluation and putting

these results into context of other models.

*We have extended this sub-subsection as follows:*

*Figure 11b shows the simulated changes in the precipitation pattern resulting from historic well-mixed greenhouse gas and solar forcing. The most important features are: the high latitudes nearly uniformly receive more precipitation; the monsoonal precipitation in North Africa and China intensifies; the ITCZ is enhanced in the western Pacific and more focused on the equator in the eastern Pacific and in the Atlantic; considerable parts of the subtropics tend to receive less precipitation. These patterns are largely consistent with precipitation changes simulated in CMIP6 models where transient aerosols are included although precipitation increases over the Indian Ocean and Northern Central Africa are not as pronounced as in the CMIP6 model mean and more pronounced over the Indonesian warm pool (compare with Figure SPM.5c in IPCC (2021)).*

Ln 398: what are spurious transformations? Water masses?

*We replaced spurious transformations with "numerical mixing in the model caused by the advection operator"*

*Spurious transformations (or mixing) refers to numerical mixing in the model caused by the advection operator. Dissipative truncation errors in horizontal advection lead to diapycnal mixing in places where isopycnals are sloping. In the ocean we use vertical advection without dissipative truncation errors. (Of course, we use FCT which can introduce additional dissipation, but we see the effect where the isopycnals are sloping, which indicates that it comes from artificial horizontal diffusion). It is generally expected that increase in horizontal resolution will lead to a reduction of numerical mixing.*

Ln 427, 432: I feel that Scafetta et al is not the best reference here. More original papers

are probably Sherwood et al (Rev. Geophys. 2020) and Meehl et al. 2020.

*We added the citation for Meehl et al. 2020*

**References**

IPCC, 2021: Summary for Policymakers. In: Climate Change 2021: The Physical Science Basis. Contribution of Working Group I
to the Sixth Assessment Report of the Intergovernmental Panel on Climate Change [Masson-Delmotte, V., P. Zhai, A. Pirani, S.L.
Connors, C. Péan, S. Berger, N. Caud, Y. Chen, L. Goldfarb, M.I. Gomis, M. Huang, K. Leitzell, E. Lonnoy, J.B.R. Matthews, T.K.
Maycock, T. Waterfield, O. Yelekçi, R. Yu, and B. Zhou (eds.)]. Cambridge University Press, Cambridge, United Kingdom and New
York, NY, USA, pp. 3−32, doi:10.1017/9781009157896.001.